# Maternal paraben exposure triggers childhood overweight development

Beate Leppert [1,13], Sandra Strunz[1,2,13], Bettina Seiwert[3,13], Linda Schlittenbauer[3], Rita Schlichting[4], Christiane Pfeiffer[1], Stefan Röder [1], Mario Bauer[1], Michael Borte[5], Gabriele I. Stangl [6,7], Torsten Schöneberg[8], Angela Schulz[8], Isabell Karkossa[9], Ulrike E. Rolle-Kampczyk [9], Loreen Thürmann [1,10,11], Martin von Bergen[9,12], Beate I. Escher[4], Kristin M. Junge[1,13], Thorsten Reemtsma [3,13], Irina Lehmann [1,10,11,13]* & Tobias Polte[1,2,13]*

Parabens are preservatives widely used in consumer products including cosmetics and food. Whether low-dose paraben exposure may cause adverse health effects has been discussed controversially in recent years. Here we investigate the effect of prenatal paraben exposure on childhood overweight by combining epidemiological data from a mother–child cohort with experimental approaches. Mothers reporting the use of paraben-containing cosmetic products have elevated urinary paraben concentrations. For butyl paraben (BuP) a positive association is observed to overweight within the first eight years of life with a stronger trend in girls. Consistently, maternal BuP exposure of mice induces a higher food intake and weight gain in female offspring. The effect is accompanied by an epigenetic modification in the neuronal Pro-opiomelanocortin (POMC) enhancer 1 leading to a reduced hypothalamic POMC expression. Here we report that maternal paraben exposure may contribute to childhood overweight development by altered POMC-mediated neuronal appetite regulation.

[1] Department for Environmental Immunology, Helmholtz Centre for Environmental Research—UFZ, Leipzig, Germany. [2] Department of Dermatology Venerology and Allergology, Leipzig University Medical Center, Leipzig, Germany. [3] Department for Analytical Chemistry, Helmholtz Centre for Environmental Research—UFZ, Leipzig, Germany. [4] Department for Cell Toxicology, Helmholtz Centre for Environmental Research—UFZ, Leipzig, Germany. [5] Children's Hospital, Municipal Hospital St. Georg, Leipzig, Germany. [6] Institute of Agriculture and Nutritional Sciences, Martin Luther University Halle-Wittenberg, Halle (Saale), Germany. [7] Competence Cluster for Nutrition and Cardiovascular Health (nutriCARD) Halle-Jena, Leipzig, Germany. [8] Medical Faculty, Rudolf Schönheimer Institute of Biochemistry, University of Leipzig, Leipzig, Germany. [9] Department Molecular Systems Biology, Helmholtz Centre for Environmental Research—UFZ, Leipzig, Germany. [10] Environmental Epigenetics and Lung Research Group, Charité—Universitätsmedizin Berlin, Berlin, Germany. [11] Molecular Epidemiology, Berlin Institute of Health (BIH), Berlin, Germany. [12] Faculty of Life Sciences, Institute of Biochemistry, University of Leipzig, Leipzig, Germany. [13] These authors contributed equally: Beate Leppert, Sandra Strunz, Bettina Seiwert, Kristin Junge, Thorsten Reemtsma, Irina Lehmann, Tobias Polte. *email: irina.lehmann@charite.de; tobias.polte@ufz.de

Childhood obesity has reached epidemic dimension in most of the developed countries and continues to increase globally. In accordance to recent reports overweight and obesity affect up to one third of children in Europe and in Northern America[1,2]. Both, life style factors such as a high caloric food intake and a predominantly sedentary behaviour, as well as genetic predisposition contribute to the risk for overweight and obesity. However, both factors alone cannot explain the fast increase in obesity rates all over the world[3]. Additional priming for overweight development by environmental factors gained increasing attention in the scientific community[4,5].

In critical time windows such as the foetal development perturbations of the physiological endocrine and metabolic signalling can result in long lasting health effects[6–8]. Synthetic chemicals interfering with the endocrine system (endocrine disrupting chemicals, EDC) represent a classical example for environmental factors contributing to programming towards overweight and obesity, in particular in the perinatal period[9]. EDCs are added to products as preservatives or plasticizers, for antimicrobial and antifungal function or as flame retardants[10,11]. Thus, exposure to EDCs is ubiquitous. These chemicals can enter the body via food and water intake, skin absorption, and inhalation[5,12]. Many of them are able to cross the blood-placenta barrier with the risk to exert their harmful properties already during prenatal development[13–15]. A group of EDCs that is suspected to interfere with ontogenesis and exert potential endocrine disrupting properties but has not yet been extensively studied in the context of overweight development are parabens.

Parabens are alkyl esters of p-hydroxybenzoic acid (PHBA) with antimicrobial and antifungal properties frequently used as preservatives in cosmetic products, toiletries, food (E214-219), and pharmaceuticals. They are found in the majority of leave-on cosmetics and in rinse-off products[16]. Parabens enter the human body mainly through ingestion or skin absorption and can commonly be detected in urine, blood and breast milk[17,18]. Paraben level vary among the different parabens, tissue and time after topical application with previously reported concentration of $0.5\,\mu g\,l^{-1}$ butyl paraben to $43.9\,\mu g\,l^{-1}$ methyl paraben in urine samples and about 4-times lower serum level[18]. The fast metabolization of parabens in the liver (half-life in humans less than 24 h) and direct urinary clearance of conjugated metabolites may explain the higher urine concentration compared to serum. Thus, as the limit of detection is comparable in the two matrices, it is more likely to detect representative paraben concentrations in urine, rather than in serum.

There is increasing evidence from epidemiological studies that parabens might be associated with breast cancer[19] and allergy development[20]. With regard to development of overweight and obesity there is only one study showing elevated urinary levels of 3,4-dihydroxybenzoic acid (3,4-DHB), a common paraben metabolite, in obese children[21]. However, the findings are not explicit because there are other sources of 3,4-DHB like plants and many fruits[21]. Furthermore, there is no study evaluating maternal exposure to parabens during pregnancy as potential risk factor for childhood overweight development.

In this study we show evidence for a positive association between maternal urinary concentrations of butyl paraben (BuP) and childhood overweight within the first eight years of life in our prospective mother-child cohort LINA. In an in vivo mouse model, we demonstrate that maternal exposure to nBuP induces an increased food intake and weight gain in female offspring. Our data provide evidence that a neuronal dysregulation of satiety could contribute to the observed gain in body weight mediated by an epigenetic silencing and reduced hypothalamic expression of the gene *proopiomelanocortin (pomc)* well-known to be involved in appetite regulation[22].

## Results

**Cosmetic products as source of paraben exposure.** Within the LINA mother-child study 629 mother-child pairs were recruited between 2006 and 2008. General characteristics of the study participants are shown in Supplementary Table 1 with no differences compared to the analysed sub-cohort for longitudinal BMI development (year 2–8) and paraben exposure; $n = 223$. Maternal paraben exposure was assessed by urine measurements showing high exposure levels for methyl paraben and lower for butyl parabens (Supplementary Table 2). As a potential source of paraben exposure the usage of cosmetic products during pregnancy was assessed by questionnaires. Indicated cosmetic products were searched for their paraben content with the TOXFOX app (described in Methods) and categorised in leave-on and rinse-off products. Since exposure time is obviously very short for rinse-off products and TOXFOX indicated only 31 out of 414 rinse-off products as paraben containing (Supplementary Fig. 1), only leave-on products with a high exposure time and high body area coverage were considered for comparison with urine measurements. Valid data on the application of paraben containing or paraben-free leave-on products during pregnancy was available from 414 participants. Indeed, 26% used at least one cosmetic leave-on product that contained parabens, whereas 56% used paraben-free leave-on products only (Supplementary Fig. 1).

When comparing measured urinary paraben concentrations, mothers that used paraben containing leave-on products had up to 3.0-fold higher concentrations for methyl paraben (MeP), ethyl paraben (EtP), n-propyl paraben (nPrP), and n-butyl paraben (nBuP) compared to mothers using paraben free leave-on products, as shown in Table 1. Thus, the usage of paraben containing leave-on products can be considered as one potential source for maternal paraben exposure.

**Prenatal paraben exposure and weight development in LINA.** Further, we defined overweight children using age and sex specific cut-offs from the International Obesity Task Force (IOTF).

**Table 1 Maternal urinary paraben concentrations [μg/l] in regard to cosmetic product application (A) usage of leave-on products during pregnancy without parabens ($n = 276$) or (B) usage of leave-on products during pregnancy containing parabens ($n = 138$).**

|  | (A) | Median | <25% | >75% | (B) | Median | <25% | >75% | P-value |
|---|---|---|---|---|---|---|---|---|---|
| MeP |  | 28.05 | 6.62 | 133.20 |  | 68.80 | 17.00 | 167.20 | 0.0018 |
| EtP |  | 1.89 | 0.51 | 9.88 |  | 2.90 | 0.79 | 18.10 | 0.0466 |
| nPrP |  | 3.20 | 0.72 | 14.60 |  | 7.40 | 1.50 | 25.80 | 0.0025 |
| iBuP |  | 0.10 | 0.05 | 0.70 |  | 0.20 | 0.05 | 0.94 | 0.1875 |
| nBuP |  | 0.41 | 0.10 | 2.70 |  | 1.24 | 0.30 | 4.55 | 0.0004 |

Shown are median, as well as first (<25%) and fourth quartile (>75%), P-values are derived from Mann–Whitney-U-test

**Table 2 Prenatal paraben exposure and weight development of children.**

| | birth | OR | 95% CI | P-value | 2–8 years | OR | 95% CI | P-value |
|---|---|---|---|---|---|---|---|---|
| MeP | | 1.18 | 0.63–2.23 | 0.606 | | 1.15 | 0.58–2.28 | 0.691 |
| EtP | | 1.39 | 0.67–2.88 | 0.382 | | 1.00 | 0.49–2.03 | 0.991 |
| nPrP | | 0.86 | 0.44–1.69 | 0.671 | | 1.04 | 0.53–2.03 | 0.920 |
| iBuP | | 1.61 | 0.81–3.18 | 0.175 | | 2.40 | 1.16–4.98 | 0.018 |
| nBuP | | 1.45 | 0.73–2.89 | 0.293 | | 2.17 | 1.06–4.47 | 0.035 |

Shown are odds ratios (OR) for macrosomia at birth ($n = 496$) and ever overweight development in early to mid childhood (age 2–8 years, $n = 223$) with 95% confidence intervals (95% CI) derived from logistic regression models adjusted for the sex of the child, smoking during pregnancy, parental school education, gestational week at delivery, existence of siblings, breast-feeding duration (not for models with only birth weight as outcome) and age of the mother at birth. OR are shown for high (3rd tertile) paraben exposure as compared to low (1st tertile) exposure in pregnancy

**Table 3 Longitudinal effect of prenatal paraben exposure on weight development in childhood (1–8 years).**

| | Tertile | All: beta | 95% CI | P-value | Female: beta | 95% CI | P-value | Male: beta | 95% CI | P-value |
|---|---|---|---|---|---|---|---|---|---|---|
| MeP | 2 | −0.10 | −0.36, 0.16 | 0.436 | −0.06 | −0.44, 0.33 | 0.77 | −0.19 | −0.53, 0.15 | 0.265 |
| | 3 | −0.04 | −0.30, 0.22 | 0.758 | −0.03 | −0.42, 0.35 | 0.87 | −0.02 | −0.35, 0.31 | 0.901 |
| EtP | 2 | 0.18 | −0.08, 0.44 | 0.183 | 0.17 | −0.26, 0.56 | 0.404 | 0.21 | −0.14, 0.56 | 0.246 |
| | 3 | −0.01 | −0.27, 0.25 | 0.944 | 0.05 | −0.36, 0.46 | 0.852 | −0.02 | −0.34, 0.31 | 0.924 |
| nPrP | 2 | 0.08 | −0.18, 0.34 | 0.555 | 0.24 | −0.14, 0.62 | 0.216 | −0.11 | −0.46, 0.25 | 0.555 |
| | 3 | −0.04 | −0.30, 0.21 | 0.729 | 0.08 | −0.21, 0.47 | 0.682 | −0.11 | −0.43, 0.22 | 0.515 |
| iBuP | 2 | 0.35 | 0.08, 0.62 | 0.011 | 0.46 | 0.07, 0.85 | 0.021 | 0.26 | −0.10, 0.62 | 0.162 |
| | 3 | 0.26 | 0.02, 0.05 | 0.035 | 0.53 | 0.16, 0.89 | 0.005 | 0.05 | −0.26, 0.36 | 0.752 |
| nBuP | 2 | 0.13 | −0.12, 0.39 | 0.342 | 0.21 | −0.17, 0.59 | 0.275 | 0.07 | −0.28, 0.42 | 0.706 |
| | 3 | 0.21 | −0.04, 0.47 | 0.095 | 0.36 | −0.03, 0.74 | 0.069 | 0.07 | −0.26, 0.40 | 0.672 |

Shown are changes in the intercept as betas with 95% confidence intervals (CI) for a change of BMI per tertile increase in paraben exposure for all children ($n = 392$), female only ($n = 193$) and male only ($n = 199$) derived from an adjusted GEE model with fitted cubic splines at ages 12, 25, 38, 61, 98 moths and exchangeable correlation matrix. Models have been adjusted for sex of the child, smoking during pregnancy, parental school education, gestational week at delivery, existence of siblings, breast-feeding duration and age of the mother at birth

Adjusted logistic regression models were applied to study a potential risk increase resulting from prenatal paraben exposure comparing weight of children at birth (>4000 g or macrosomia vs. <4000 g) and afterwards (overweight vs. non-overweight children; Table 2). Since reference data are not available for one-year-old children, only the time window between age two and eight could be considered for this analysis. While high exposure to parabens did not affect the risk for macrosomia at birth, there was evidence for the long chain parabens iBuP and nBuP to increase the risk for ever overweight in early to mid-childhood (Table 2).

To investigate the impact of prenatal paraben exposure on longitudinal children's weight development we applied GEE analysis on BMI-data from age 1–8 years. We found evidence for an early manifestation of differences between prenatally low (1st tertile) and high (3rd tertile) exposed children for iBuP and nBuP (intercept: $0.26 \, \text{kg m}^{-2}$, 95% CI: 0.02,0.05; $0.21 \, \text{kg m}^{-2}$, 95% CI: −0.04,0.47, respectively) and no evidence for MeP, EtP and nPrP (Table 3). Thereby, the BuP (both iBuP and nBuP) related BMI increase was more evident in girls compared to boys and effects hold true after adjustment for confounders. There was no evidence for a change of the slope between exposure groups over time.

**Impact of paraben exposure on adipocyte development in vitro**. Parabens have been shown to promote adipocyte differentiation in mouse fibroblasts[23]. To evaluate a potential direct effect of parabens on human adipocyte differentiation an established human mesenchymal stem cell differentiation assay was applied[9]. Adipocyte differentiation, assessed by the cell index values of impedance-based real-time monitoring and by the amount of triglyceride storage, was not affected by nBuP exposure (Fig. 1a–c) nor by the other parabens measured in LINA (Supplementary Fig. 2A, B). Furthermore, gene expression of the transcription factor *PPARG* showed no differences in nBuP-treated cells compared to control (Fig. 1d). Looking closer into PPARγ regulation, we found no evidence for PPARγ activation by nBuP in an artificial reporter gene assay (Supplementary Table 3). Moreover, nBuP exposure did not activate the androgen, progesterone and glucocorticoid receptor but exerted a strong impact on oestrogen receptor-α (ER-α) activity (Supplementary Table 3). Interestingly and in contrast to the other results, *leptin (LEP)* expression in adipocytes was downregulated by nBuP with a significant effect even at 0.5 μM. For validation of these findings the secretion of adiponectin and leptin into the cell culture supernatant was assessed. Also decreased levels of secreted leptin were approved after exposure to nBuP, with a significantly effect at 10 μM. Secreted adiponectin levels significantly increased after exposure to nBuP in a concentration-dependent manner (Fig. 1e). Paraben exposure at the used concentrations had no effect on cell viability (Supplementary Fig. 2C).

Furthermore, using adipocytes differentiated from primary mouse mesenchymal stem cells we observed comparable results to the human in vitro analyses showing no direct effect of nBuP exposure on adipogenesis (Supplementary Fig. 3A–C).

**Maternal nBuP increased weight in offspring of female mice**. To further investigate a potential causal relationship between an increased risk for childhood overweight and maternal exposure to certain parabens, we exposed Balb/c mice during gravity and the breast-feeding period to BuP and measured offspring's body weight and several metabolic parameters. Due to its higher human exposure (see Supplementary Table 2) nBuP (1.75 μg twice per week s.c.) was used in the experimental setup. nBuP urine concentration was measured in exposed mice and was found to be in a comparable range as observed in the LINA study (highest tertile urine concentration LINA mothers: $18.5 \, \mu\text{g l}^{-1}$; mean urine concentration in exposed mice: $19.6 \, \mu\text{g l}^{-1}$).

Female offspring from perinatally nBuP-exposed dams showed a significantly higher weight than control animals over the entire observation period with a weight increase of 20 to 45% compared to 10% in the controls (GEE estimate: 3 g higher body weight in the exposed group; SE: 0.5 g, $p = 5.3 \times 10^{-9}$, Fig. 2a). The elevated body

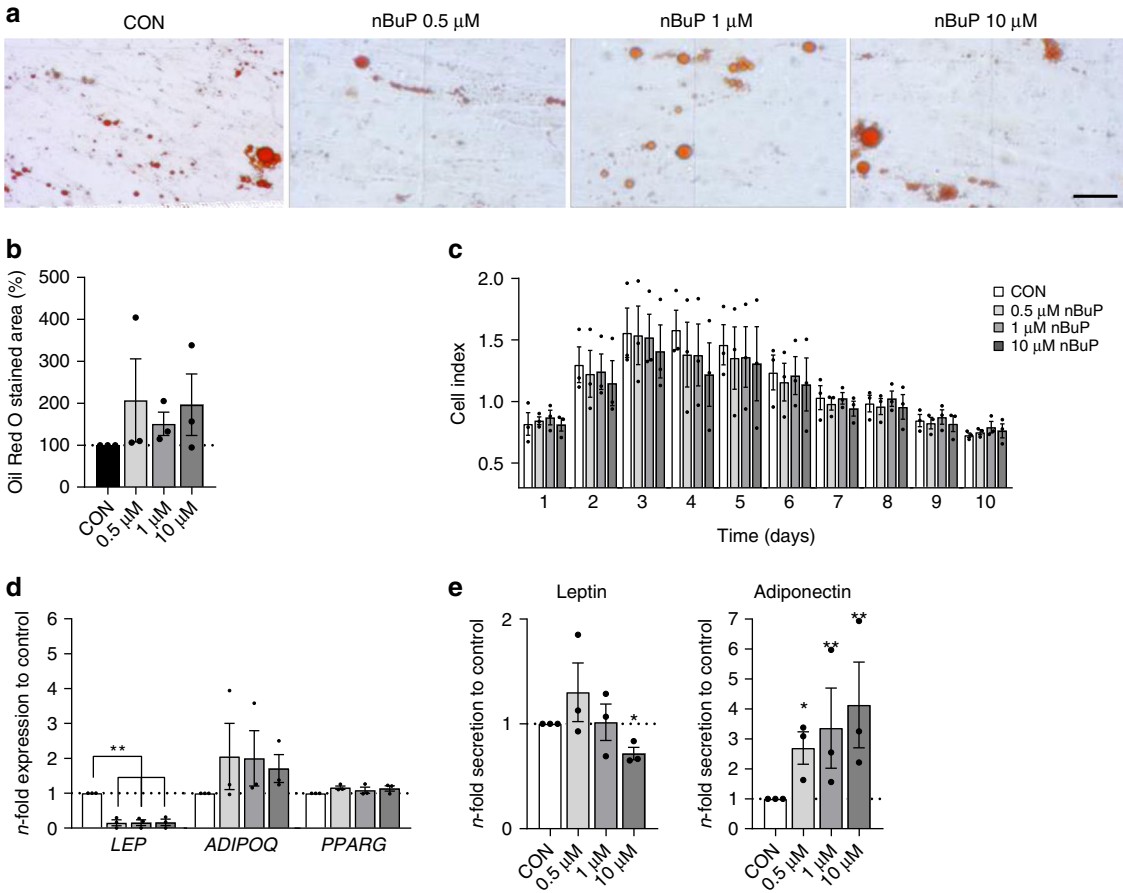

**Fig. 1 Effect of nBuP exposure on adipocyte differentiation. In vitro adipocyte differentiation from human MSCs in the presence of nBuP. a**
Representative Oil Red O stained pictures after differentiation (scale bar: 100 μm). **b** Triglyceride storage of adipocytes assessed via Oil Red O staining. **c**
Real-time monitoring of cell differentiation (xCELLigence: normalised cell index) over a 17-day period. **d** Gene expression of *leptin (LEP)*, *adiponectin*
*(ADIPOQ)* and the transcription factors *peroxisome proliferator-activated receptor gamma (PPARG)*. **e** Leptin and adiponectin levels in cell culture supernatant.
Data are expressed as mean ± SEM of $n = 3$ experiments. Significant differences were derived by ANOVA with *$P < 0.05$ and **$P < 0.01$. Source data are
provided as a Source Data file.

weight became manifest shortly after birth and was linked to a
higher fat and a lower lean mass as measured by whole body
composition analysis using nuclear magnetic resonance technology
(Fig. 2b). In contrast, weight development of male offspring was not
affected by perinatal nBuP exposure (GEE estimate: 0.5 g higher
body weight in the exposed group, SE: 0.6 g, $p = 0.34$, Fig. 2a, b).
Furthermore, female mice from nBuP-exposed dams showed an
increase in weekly food intake over the observation period
compared to control mice (GEE estimate: 0.5 g additional food
intake per week in the exposure group; SE: 0.2 g, $p = 0.006$; Fig. 2c).
In addition, while the fasting serum glucose levels were elevated in
the female offspring of nBuP-exposed dams, we did not detect
impaired glucose and insulin tolerances (Fig. 2d, e). The male
progeny showed no differences in glucose and insulin levels
compared to the control mice from unexposed dams (Fig. 2d, e).
Further, nBuP exposure to female adult mice had no effect on
weight, food intake and leptin serum levels (Supplementary Fig. 4).

Within the female F1 generation, elevated body weight of mice
was associated with an increased size of adipocytes (Fig. 3a, b).
Gene expression analysis revealed no changes for the *glycose
transporter 4 (glut4)*, the *insulin receptor (insr)* and *pparg*
in adipose tissue of female offspring from nBuP-exposed dams
compared to control animals (Fig. 3c). Furthermore, while
serum leptin levels were elevated in the offspring from nBuP-
exposed dams, the concentrations of adiponectin, resistin, ghrelin,
and insulin were not affected compared to control animals

(Fig. 3d). In addition, maternal nBuP exposure did not
influence 17β estradiol levels in female and male offspring
(Supplementary Fig. 5).

**Maternal nBuP reduced POMC expression via nPE1 methyla-
tion.** The results from the human adipocyte differentiation assay
revealed no conclusive impact of nBuP exposure on adipogenesis.
Moreover, the female offspring of nBuP-exposed dams released
leptin as expected as a result of a higher weight gain. Accordingly,
we next focused on gene analyses important for the central reg-
ulation of food intake in the hypothalamus. Expression of the
*leptin receptor (lepr)* mRNA was downregulated in female off-
spring of nBuP-exposed dams (Fig. 4a) suggesting a potentially
impaired leptin signalling. This finding was supported by a very
low expression of *pomc* mRNA in female offspring compared to
the progeny from non-exposed mice (Fig. 4a). The mRNAs of the
*melanocortin type 4 receptor (mc4r)*, a target for the POMC
cleavage product alpha-melanocyte-stimulating hormone (α-
MSH), the *agouti-related neuropeptide gene (agrp)* and the *insulin
receptor X (insr)* were unaffected in the overweight mice (Fig. 4a).

Because environmental factors influencing ontogenesis that
may affect the disease risk later in life have been shown to exert
their effects via the induction of epigenetic changes such as DNA
methylation[24,25], we investigated whether the paraben-driven
*pomc* downregulation in female offspring is due to nBuP-induced
alterations in DNA methylation of regulatory regions (nPE1,

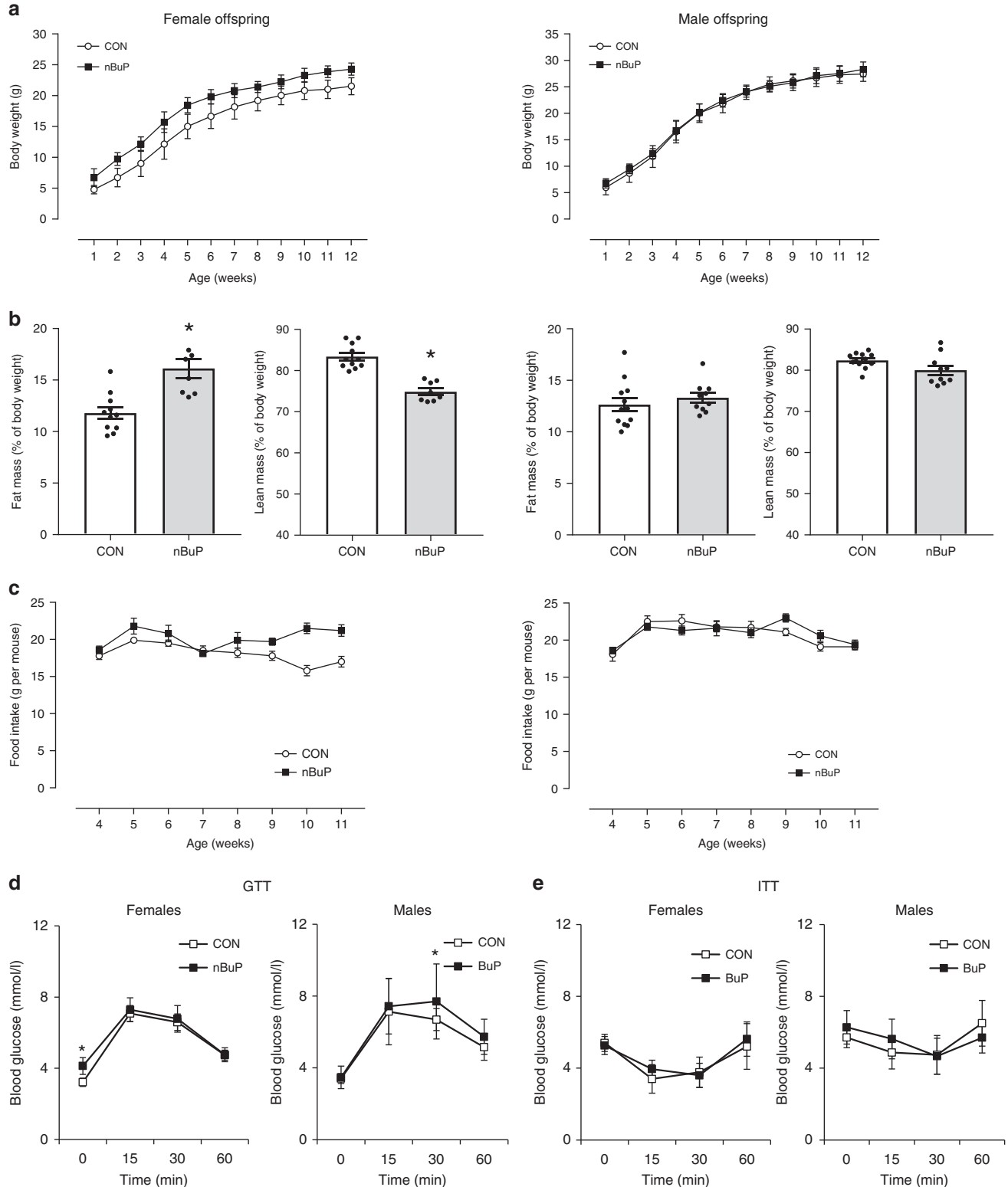

**Fig. 2 Perinatal exposure to nBuP and weight development, food intake and glycose metabolism in the offspring. a** Bodyweight development, (CON: $n = 18$, nBuP: $n = 16$), **b** body composition (CON: $n = 11$, nBuP: $n = 9$) and **c** food intake (CON: $n = 10$, nBuP: $n = 6$) are shown from female (left) and male offspring (right) from nBuP-exposed dams, **d** Glucose tolerance test (GTT, CON: $n = 18$, nBuP: $n = 10$) and **e** insulin tolerance test (ITT, CON: $n = 15$, nBuP: $n = 10$) were performed in 9-weeks old offspring. Data are expressed as mean ± SEM, *$P < 0.05$, unpaired $t$-test. For longitudinal analysis GEE models were applied (**a** female mice $\beta = 3$ g, SE = 0.5 g, $p = 5.3 \times 10^{-9}$, male mice $\beta = 0.5$ g, SE = 0.6 g, $p = 0.34$; **c** female mice $\beta = 0.5$ g per week SE = 0.2 g, $p = 0.006$, male mice $\beta = 0.25$ g per week, SE = 0.18, $p = 0.17$). Source data are provided as a Source Data file.

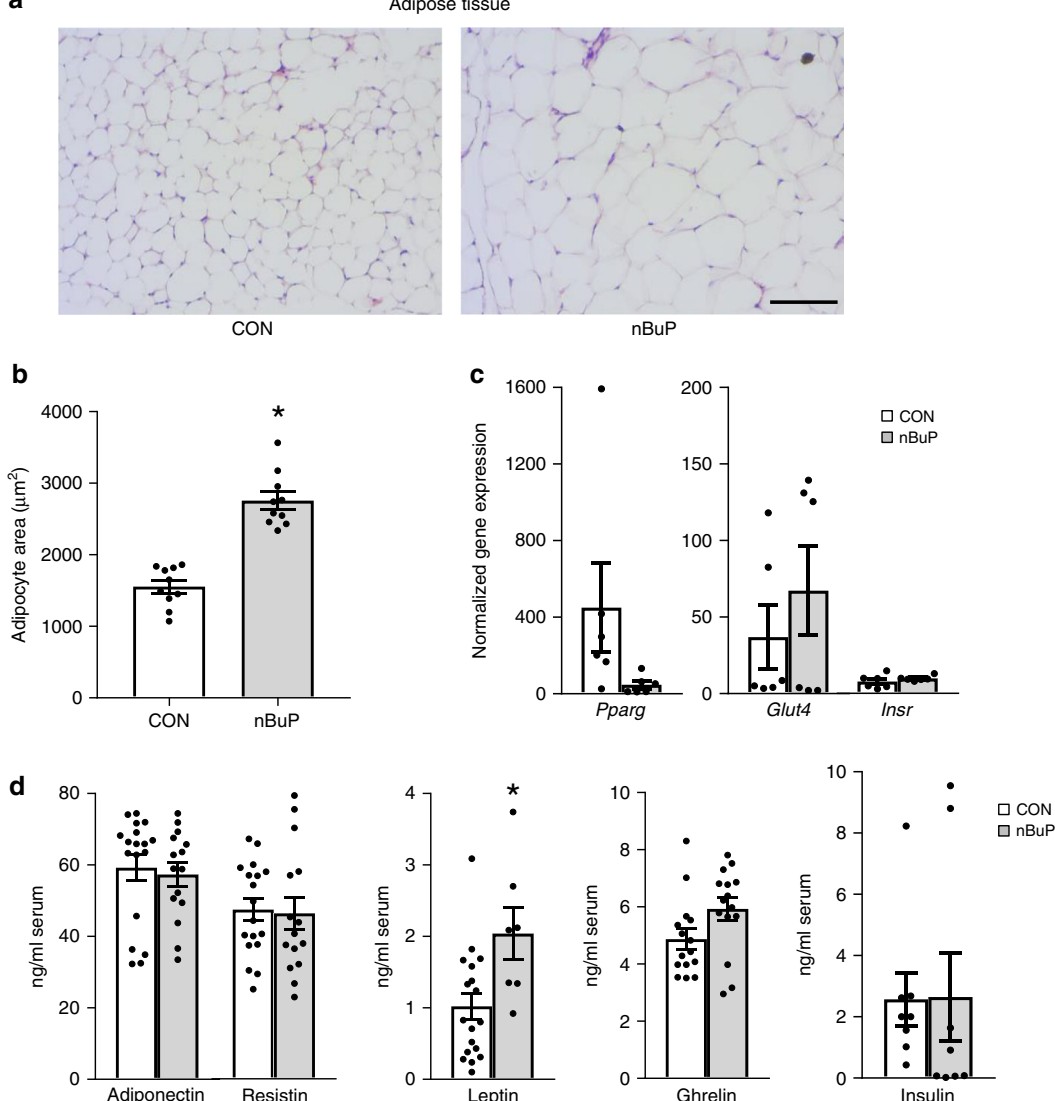

**Fig. 3 Perinatal nBuP exposure, adipocyte area and gene and protein expression of key genes in the offspring. a** Representative picture of stained slices (H&E, ×20, scale bar: 100 μm) of visceral adipose tissue and (**b**) the illustration of the adipocyte area from female offspring of nBuP-exposed dams ($n = 10$), (**c**) The mRNA expression levels of selected target genes investigated in adipose tissue of female offspring ($n = 6$), (**d**) Levels of adiponectin and resistin (CON: $n = 18$, nBuP: $n = 15$), leptin (CON: $n = 18$, nBuP: $n = 7$), ghrelin ($n = 15$) and insulin ($n = 8$) measured in serum of female offspring. Data are expressed as mean ± SEM, *$P < 0.05$, unpaired $t$-test. Source data are provided as a Source Data file.

nPE2, Supplementary Fig. 6) of the *pomc* gene. We detected an increased DNA methylation of nPE1 (Fig. 4b) while we did not observe any methylation changes in promoter and nPE2 regions (Supplementary Fig. 7). Furthermore, the hypermethylated nPE1 and reduced *pomc* mRNA expression was already detectable in the offspring from nBuP-exposed dams directly after weaning (Fig. 4c).

To evaluate whether the nBuP-induced hypermethylation is linked to overweight development in the offspring, one-week-old pups from nBuP-exposed dams were treated with the DNA methyltransferase inhibitor 5-Aza-2'-deoxycytidine (Aza) for two weeks until weaning[26]. Treatment of the offspring with Aza reduced the body weight and the food intake caused by maternal nBuP exposure (Fig. 4d), as well as adipocyte area, and leptin serum levels and restored *lepr* expression in the hypothalamus (Supplementary Fig. 8). Moreover, the paraben-induced nPE1 hypermethylation and the diminished hypothalamic *pomc* expression in the offspring from nBuP-exposed dams were

reversed in the presence of Aza (Fig. 4e, f), while nPE1 methylation and *pomc* mRNA expression in the control group were reduced.

## Discussion

In this study a complex translational research design was applied to study the potential impact of maternal paraben exposure on children's weight development. Findings from the prospective birth cohort study LINA were used to establish hypothesis driven mechanistic studies in human and mouse in vitro models. Based on these results a mouse in vivo experiment provided the final proof-of-concept and deeper mechanistic insight (Fig. 5).

Evidence from our LINA cohort study indicates that maternal exposure to butyl paraben (BuP) may trigger overweight development in early childhood. Using a mouse model, we verified these epidemiological data and provide a mechanistic explanation by demonstrating that prenatal exposure to nBuP induced an

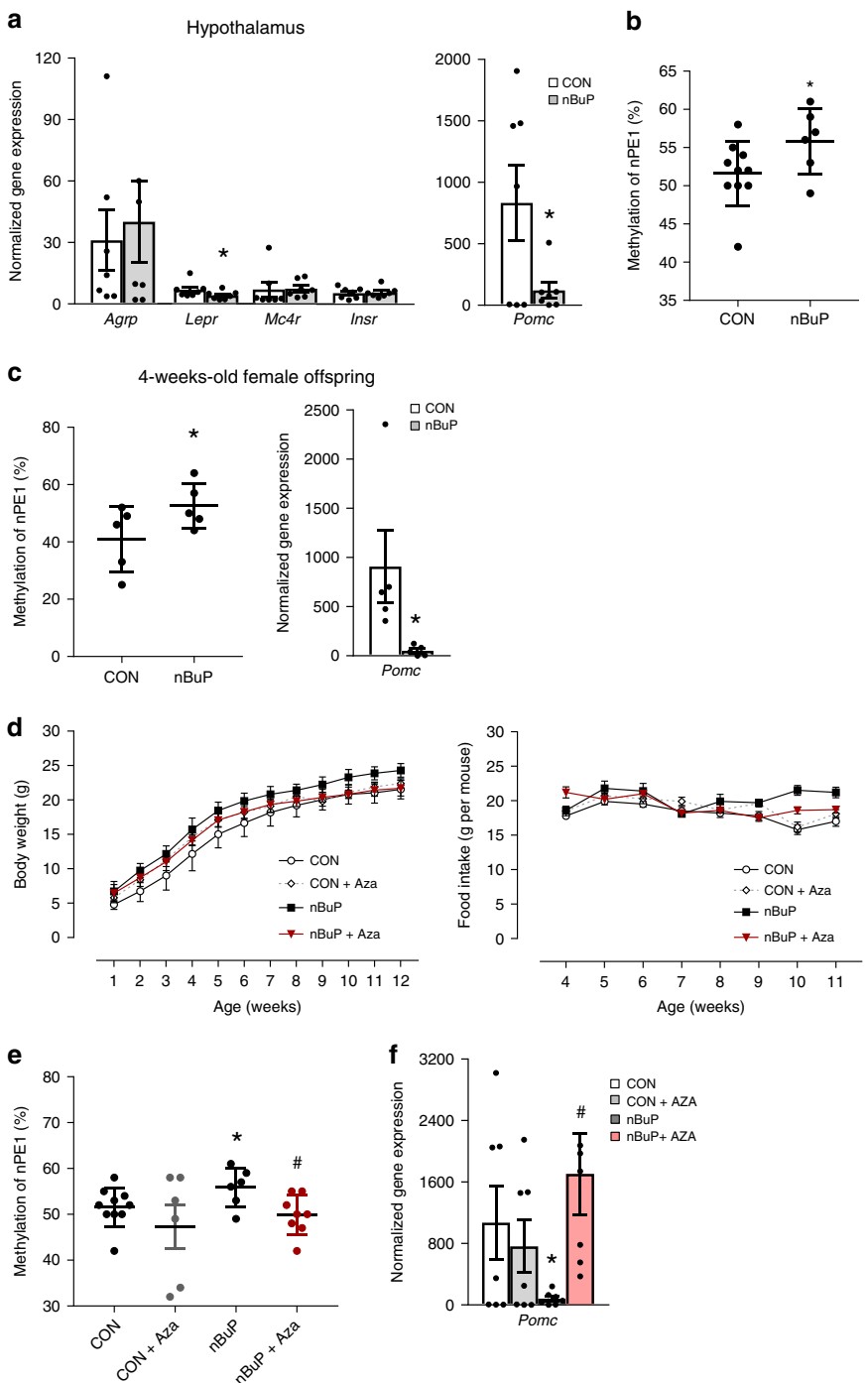

**Fig. 4 Perinatal nBuP exposure reduced *pro-opiomelanocortin (pomc)* expression and induced a DNA hypermethylation of nPE1. a** Expression levels of genes important for the neuronal regulation of satiety and hunger ($n = 7$) and (**b**) DNA methylation of the neuronal POMC enhancer nPE1 (chr12:3941750-3942350, CON: $n = 10$, nBuP: $n = 6$) were analysed in the hypothalamus of the 12-week-old female offspring. Values for DNA methylation in both groups are pictured for CpG1 (cg3942264). **c** DNA Methylation of nPE1 and *pomc* gene expression from the 4-weeks-old female offspring of nBuP-exposed dams are shown ($n = 5$). **d** After treatment of F1 mice with the DNA methyl-transferase inhibitor Aza body weight development (CON: $n = 18$, CON + Aza: $n = 12$, nBuP: $n = 16$, nBuP + Aza: $n = 8$) and food intake (CON: $n = 10$, CON + Aza: $n = 9$, nBuP: $n = 6$, nBuP + Aza: $n = 8$) were evaluated. **e** nPE1 methylation (CON: $n = 10$, CON + Aza: $n = 6$, nBuP: $n = 6$, nBuP + Aza: $n = 8$) and (**f**) *pomc* gene expression ($n = 7$) was measured in Aza-treated female offspring from nBuP-exposed dams. Data are expressed as mean ± SEM, *$P < 0.05$ for nBuP vs. CON, #$P < 0.05$ for nBuP + Aza vs. nBuP, Student's unpaired *t*-test. For longitudinal analysis (**d**) GEE models were applied (see Supplementary Table 5). Source data are provided as a Source Data file.

increased food intake and weight gain in female offspring. Our data indicate that a neuronal dysregulation of food intake could contribute to the observed effect since we could show an epigenetic silencing and reduced hypothalamic expression of the gene

*proopiomelanocortin (pomc)* known to be strongly associated with food intake regulation.

Maternal urine samples at 34th week of gestation contained detectable paraben concentrations, supporting the hypothesis of

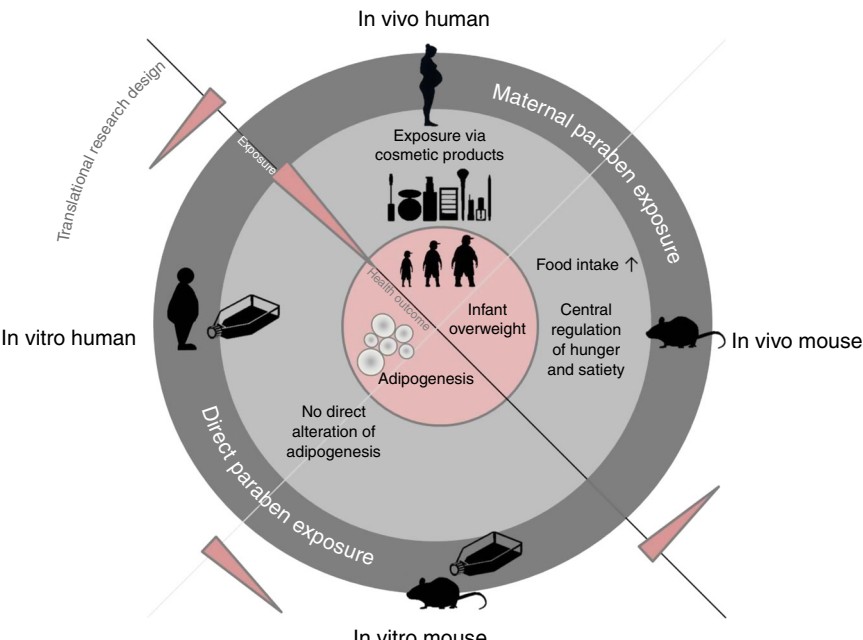

**Fig. 5 Translational research design.** Summary scheme of the present investigations describing the complex translational research design of the study. Findings from the prospective birth cohort study LINA were used to establish hypothesis driven approaches in human and mouse in vitro studies. Based on these results overall mechanistic analyses were addressed in the final mouse in vivo settings.

ubiquitous human exposure. As parabens are frequently used in cosmetic products, it seems reasonable that cosmetics are a principal source for human paraben exposure. This assumption could be supported by linking the questionnaire data on the usage of cosmetics of the mothers with urine paraben measurements within the LINA cohort. Mothers that used paraben-containing cosmetic leave-on products on a daily basis had significantly higher urinary paraben concentrations, although the actual number of paraben-containing products used during pregnancy might be underestimated due to limited data on cosmetic product composition from the LINA recruitment period between 2006 and 2008. However, despite the significant link between the usage of leave-on cosmetics and urinary paraben concentrations, we cannot exclude that further sources in addition to cosmetics have contributed to increased maternal urine concentrations. For the actual exposure situation it has to be considered that e.g., the Commission of the European Union has directed a maximum concentration for particular preservatives in cosmetic products in 2014[27]. Thus, current exposure levels might be lower compared to the LINA recruitment period.

Data from the LINA study provide some evidence that high prenatal paraben butyl paraben exposure can contribute to an increased risk of being overweight in early to mid-childhood. Stratified for sex, this effect seems to be stronger in girls compared to boys. Interestingly, effects for short chain parabens could not be confirmed in association with children's weight development.

To verify a potentially direct effect of maternal nBuP exposure on weight development and metabolism we applied an appropriate mouse model[9]. The dose of paraben exposure were estimated from a daily intake of BuP in humans of 20 µg kg⁻¹ body weight based on a No Observed Adverse Effect Level (NOAEL, 2 mg kg⁻¹ body weight) reported in a previous study[28,29]. This dose corresponds to a weekly BuP uptake of 3.5 µg in our mice experiments. The resulting urinary BuP levels in the mouse model were similar to those observed in highly exposed mothers of the LINA cohort suggesting that BuP concentrations investigated in the mouse model are representative for the real exposure situation

in humans. Our results demonstrated that female but not male offspring of exposed dams showed higher weight over the entire observation period.

To identify the underlying mechanisms of the nBuP-induced weight gain we investigated a possible effect on adipocyte differentiation using a human and a murine mesenchymal stem cell assay. We have successful applied this model earlier to test adipogenic effects of bisphenol A[9]. Here, we chose paraben concentrations based on previous publications[23] and the assumption that 0.5 µM (the lowest concentration used) are approximately equivalent to the highest paraben concentrations found in LINA for MeP. Interestingly, our results did not indicate that nBuP up to a concentration of 10 µM has a direct impact on adipogenesis. This was in contrast to previous in vitro studies demonstrating an increasing effect of nBuP exposure on adipocyte differentiation as revealed by adipocyte morphology and lipid accumulation of mouse fibroblast cells with increasing effect size from MeP to nBuP (MeP < EtP < …..nBuP)[23,30,31]. Moreover, we did not detect an activation of PPARγ by nBuP, although other studies using mouse cell lines reported an activation of this receptor which is known to be crucial in adipogenesis[23].

Although in vitro adipocyte differentiation was unaffected in our human stem cell differentiation assay, we found adiponectin and leptin expression and secretion to be altered. In vivo, leptin activates POMC neurons to induce satiety in the hypothalamic food intake control region via α-MSH release[22]. Hence, reduced leptin signalling may lead to an inappropriate satiety signalling and increased food intake, which could then promote overweight development in children.

In our mouse model we showed that the weight gain in the offspring of exposed animals was accompanied by an increased food intake, fat mass and increased adipocyte size but no differences in glucose and insulin tolerances. In vivo, leptin is expressed in adipose tissue proportionally with fat mass[32]. As expected, an increase of serum leptin level was observed in mice with a higher fat to lean mass ratio. In accordance with our in vitro analysis there was no effect on adipocyte differentiation and gene expression of *pparg* in adipose tissue. These findings

indicate that maternal nBuP exposure may not directly affect adipogenesis leading to the observed weight gain. Therefore, looking more closely on hypothalamic food intake regulation, we found prominent genes to be differentially expressed between maternally exposed and non-exposed offspring. Gene expression of the *leptin receptor* (*lepr*) and *pomc* was downregulated due to nBuP exposure. POMC is involved in anorexic signalling pathways to suppress food intake. After stimulation, α-MSH is released from POMC by proteolysis and binds to MCR4 to induce satiety[33,34]. To shed some further light on a potential mechanism, we found a DNA hypermethylation of the neuronal enhancer nPE1 but no methylation changes in the POMC promoter and nPE2 regions. It has been recently shown that nPE1 is primarily responsible for the POMC transcriptional regulation leading to a higher body weight and food intake only after deletion of this particular enhancer e.g., in comparison to nPE2[35]. Treating pups with the DNA methyltransferase inhibitor Aza reduced the nBuP-induced increased body weight, food intake and *pomc* expression suggesting a direct link between DNA hypermethylation and the observed phenotype. Results of Aza treatment experiments have always to be interpreted with caution since the Aza treatment affects the entire genome. However, comparing the Aza effect in controls and nBuP exposed animals there is some evidence for an involvement of DNA hypermethylation in phenotype development. Anyhow, further investigations are needed to clarify the epigenetic alterations more specifically. We also cannot exclude that additional pathways might be involved in mediating the nBuP-induced increased weight.

With our mouse experiment, we demonstrated that the foetal development seems to be a sensitive time window for nBuP exposure with respect to body weight regulation while exposure of the adult animals did not show an effect on weight gain, food intake and serum leptin levels.

The fact that the weight gain was preferential seen in girls or the female offspring in our experimental model and that nBuP exposure affects the ER-α activity might suggest a role of the different oestrogen levels between males and females. Although oestrogen is rather associated with an increased risk to develop visceral obesity due to its reduced production in woman after menopause[36], other studies have shown that oestrogen treatment at critical early periods of development induced an obesogenic effect which was maintained until adulthood[37,38]. Nevertheless, the research is currently lacking comprehensible mechanistic explanations for the observed associations of prenatal exposure to oestrogenic compounds and an increased risk for childhood overweight.

In summary, our study results strongly suggest that prenatal exposure to nBuP increases overweight development in the offspring. The effect is mediated by an epigenetic enhancer modification leading to an altered expression of *pomc*, which plays a crucial role in controlling the neuronal regulation of food intake. Our findings do not implicate to disregard the importance of a balanced diet or sufficient exercise for weight management but call attention to the great significance of environmental exposures during pregnancy for the disease susceptibility in later life.

## Methods

**LiNA study design and sample collection.** The German prospective birth cohort LINA (Lifestyle and environmental factors and their Influence on Newborns Allergy risk) recruited 622 mothers (629 children) at 34 weeks of gestation between May 2006 and December 2008 in Leipzig, Germany. Mothers with severe immune or infectious diseases during pregnancy were excluded from the study. Standardised self-administered questionnaires were collected annually starting in pregnancy, assessing general information about personal lifestyle, housing and environmental conditions and disease state. Study participation was voluntary and written informed consent was obtained by all participants. The study was approved by the Ethics Committee of the University of Leipzig (file ref # 046-2006, 160-2008,

160b/2008, 144-10-31052010, 113-11-18042011, 206-12-02072012, #169/13-ff, #150/14-ff). The LINA study is registered in birthcohorts.net.

**Analysis of urinary paraben concentrations in human samples.** Urinary concentrations of eight paraben species were determined in 504 maternal urine samples of the 34[th] week of gestation (reduced sample size due to available urine samples). The samples were prepared and analysed as described and validated by Schlittenbauer et al.[39]. In brief, urine samples, quality controls, and solvent blanks were thawed at room temperature, vortexed, and centrifuged. Aliquots were spiked with internal standard solution and a deconjugation standard. Hydrolysis was achieved by adding enzyme buffer solution and incubation in an ultrasonic bath. The enzyme was removed by centrifugation/filtration using Amicon® Ultra-0.5 filters. LC-MS analysis was performed on an UPLC™ system (ACQUITY I-Class, Waters Cooperation Milford, MA, USA) coupled to a triple quadrupole mass spectrometer (Xevo TQ-S, Waters Cooperation, Manchester, UK) equipped with an electrospray ionisation (ESI) source. For each analyte (MeP, EtP, iPrP, nPrP, sBuP, iBuP, nBuP, BzP), a quantification (Q) and a confirmation (q) MRM transition was selected. Quality criteria for positive confirmation of peaks were (i) presence of both MRM transitions, (ii) a retention time within 0.03 min, and (iii) a relative ion ratio within 50 to 150% compared to a standard. The limits of quantification (LOQ) were 0.5 µg l$^{-1}$ for MeP and 0.1 µg l$^{-1}$ for the others parabens (EtP, iPrP, nPrP, sBuP, iBuP, nBuP, BzP). For 498 participants valid paraben concentrations above the detection limit were obtained (cases used for weight analyses are shown in Supplementary Table 2). For iPrP, sBuP and BzP more than 70% of samples had concentrations below the LOQ and were therefore excluded from further analysis. For paraben analysis of mouse samples 20 µl urine and 180 µl milliQ water was mixed and the same protocol as described above was applied.

**Assessment of cosmetic product application.** The usage of cosmetic products during pregnancy was assessed within the recruitment period via questionnaires at the 34th week of gestation. Participants could name up to 6 cosmetic products that were used on a daily basis. These cosmetics were categorised in leave-on products (crèmes, body lotion, make-up or leave-on facial cleansings) and rinse-off products (toothpaste, rinse-off facial cleansings, hairstyling, perfume and others). The content of parabens in the named cosmetic products was assessed between April and July 2016 via the TOXFOX app for iOS from the Friends of the earth Germany (Bund für Umwelt und Naturschutz Deutschland), which provides information about the content of endocrine disrupting chemicals like parabens in an online data base of cosmetic products. Insufficiently indicated cosmetic products that could not be assigned specifically to a particular brand/product were excluded from further analysis.

**Anthropometric data collection.** Children's body weight and height were assessed annually during clinical visits or were obtained from regular preventive medical check-ups asked for in the LINA questionnaires. At birth and the one-year follow-up children's length was measured horizontally; afterwards standing height was measured to the nearest 0.1 cm. Body weight was measured to the nearest 0.1 kg. To adjust for child's age and sex, BMI was classified according to the Extended International (IOTF) Body Mass Index Cut-Offs for Thinness, Overweight and Obesity in Children[40] on a detailed monthly basis for children from age 2–8 years. Children with BMI equivalent to an adult BMI ≥ 25 (for example: 36-month-old girl: 17.64 kg m$^{-2}$, 36-month-old boy 17.85 kg m$^{-2}$) were classified as overweight (OW), children with BMI equivalent to an adult BMI < 25 as non-overweight (NOW). Further, we have classified weight at birth as ≥4000 g (or macrosomia; considering gestational age as a confounding factor in our model analyses) or as birth weight <4000 g (n = 496). Longitudinal BMI information was available for 396 children from year 1–8. Classification in NOW and OW was performed in early to mid-childhood (year 2–8; n = 223). Due to missing reference data the first year could not be included in this analysis. This should not be of relevance for the paraben effect since Körner et al.[41] described a longitudinal overweight stabilising effect only after the age of 3 years. We were able to verify the OW classification based on a Bioelectrical Impedance Analysis (BIA) performed at children's age of eight. Children classified as OW had about 30% higher body fat mass percentage at 8 years of age compared to normal weight children (fat mass overweight = 19.3%, n = 58; fat mass normal weight = 14.9%, n = 138, P < 0.001).

**In vitro adipocyte differentiation.** Human adipose-derived mesenchymal stem cells (MSC) derived by a female donor (ATCC, LGC Standards (Wesel, Germany), PCS-500-011; LOT #59753760) and appropriate culture media were purchased from ATCC (LGC Standards,Wesel, Germany, PCS-500-030, PCS-500-040 and PCS-500-050). Cells were cultured under normal conditions at 37 °C, 5% $CO_2$ and 95% humidity according to the manufacturer's instructions[9]. In brief, for adipocyte differentiation MSCs passage 3–5 were seeded at 9.000 cells per cm$^2$ in a 96-well plate and grown to 70% confluence. For initiation of differentiation cells were fed with Adipocyte Differentiation Initiation Medium (ADIM; ATCC Adipocyte Differentiation Toolkit PCS-500-050). ADIM was changed to Adipocyte Maintenance Medium (ADMM; ATCC Adipocyte Differentiation Toolkit PCS-500-050) after 4 days and changed regularly. Cells were treated with parabens at different concentrations (working solution in 0.05% ethanol) during the entire differentiation

period. Butyl paraben exposure was represented in the experimental setup by nBuP due to its higher human exposure (see Supplementary Table 1) compared to iBuP. Proliferation and differentiation were monitored in real-time by the impedance-based xCELLigence MP System (ACEA Biosciences Inc., San Diego, USA) on a microelectrode 96 well E-View-Plate (ACEA Biosciences Inc.). A cell index was assessed every 10 min by normalising an electrical impedance measurement to a bland value for each well. Measurements were paused for media changes. After a total of 16 days, cells were stained with Oil Red O for triglyceride depots and harvested for qPCR analysis. Oil Red O staining was quantified via absorbance measurement at 510 nm. qPCR was carried out in a Roche Light Cycler 480 system with primer sequences being listed in the Supplementary Table 4. Human adiponectin and leptin protein concentrations were assessed in cell culture supernatants of human MSC derived adipocytes by commercially available ELISA kits (Adiponectin ELISA Kit, Human (KHP0041); Leptin ELISA Kit, Human (KAC2281) from Invitrogen/Thermo Fischer, Ulm, Germany) according to the manufacturer's instructions. Undiluted samples were measured in triplicates, using a 1× secondary antibody and compared to 8 standards, ranging from 0–1000 pg ml$^{-1}$ for leptin and 0–23 ng ml$^{-1}$ for adiponectin. Cytotoxicity of exposed chemicals was assessed prior to and after adipocyte differentiation via a MTT (3-(4,5-dimethylthiazol-2-yl)-2,5-diphenyoltetrazoliumbromid) assay.

To follow our translational research design, a validation of the human adipose-derived MSC experiment was performed using adipocyte derived murine MSCs (Cyagen, MUBMD-01001, Strain C57BL/6 Mouse Adipose-Derived Mesenchymal Stem Cells) according to manufacturer's instructions. In brief, for adipogenic differentiation mMSC were passaged onto 12-well plates at a concentration of 20,000 cells per cm$^2$ and were grown to at least 100% confluence in Basal Media (Cyagen, MUXMX-90011). After that the cells underwent adipogenic differentiation according to manufacturer's instructions (using Cyagen GUXMX-90031) following a maximum of 5 cycles of 3 day induction and 1 day maintenance period. With each cycle of induction, cells were exposed to Ethanol control (0.05%), 0.5 µM, 1 µM, or 10 µM nBuP. Rosiglitazone provided by Cyagen was only used as positive control. After the cells formed big and round lipid droplets they were harvested and processed for further analyses (see human MSCs for Oil Red O staining and qPCR analyses).

**Reporter gene assay.** Butyl paraben was tested in five reporter gene cell lines[42] based on the human nuclear receptors peroxisome proliferator activated receptor PPARγ (CellSensor PPARγ-UAS-BLA293-H, Thermo Fisher Scientific), androgen receptor AR (CellSensor AR UAS BLA GRIPTITE), oestrogen receptor ER (CellSensor ER a UAS BLA GRIPTITE), progesterone receptor PR (CellSensor PR-UAS-BLA HEK293T), and glucocorticoid receptor GR (CellSensor GR-UAS-BLA HEK293T). The experiments were performed as described previously[43,44] using in parallel reference compounds for quality control and dosing the parabens in methanol assuring that the final methanol was <0.1%. 4000 cells (PPARγ), 3500 cells (ER), or 4500 cells (AR, PR, GR) were seeded per well in 384-well plates with 30 µl of medium and incubated for 24 h at 37 °C and 5% CO$_2$. The methanolic stock solutions of the paraben were diluted with assay medium and a 10-step serial dilution-series was prepared in a 96-well dosing plate and 10 µl of each dilution were added to the 384-well cell plates. The dosed cells were incubated another 22 h at 37 °C and 5% CO$_2$. Activation of nuclear receptors PPAR, AR, ER, PR, GR was detected after adding 8 µl of FRET detection including ToxBlazer mixture per well followed by 2 h incubation at room temperature. Blue (activated) and green signals (inactive) were detected using an excitation filter of 409 nm and emission filters of 460 nm and 530 nm, respectively. Induction of the transcription factor or, more precisely, expression of the reporter gene β-lactamase was measured by FRET fluorescence. For the cytotoxicity, an excitation filter of 600 nm and an emission filter of 665 nm were used. The activation of ARE was detected by quantification of the luciferase reporter product using luciferin/ATP reagent as described previously[43,45]. Concentration-response curves were plotted for cytotoxicity and activation of the reporters. The inhibitory concentration for 10% of cytotoxicity IC$_{10}$ was deduced from a log-sigmoidal fit of the concentration-cytotoxicity curve[43]. Only concentrations below IC$_{10}$ were further processed for activation. Linear concentration-response curves up to 30% effect level or an induction ratio of 4 for the AREc32 assay were plotted and effect concentration triggering 10% of the maximum effect of a positive control, EC10, or causing an induction ratio IR of 1.5, EC$_{IR1.5}$ were derived[46].

**Mice.** BALB/cByJ mice (6–8 weeks of age) were obtained from the Elevage Janvier Laboratory (Le Genest St Isle, France). Mice were bred and maintained in the animal facility at the University of Leipzig (Germany) under conventional conditions with 23 °C room temperature, 60% humidity, and 12 h day/night rhythm. Control and nBuP-exposed dams and pups were housed in polyphenylsulfone (PPS) cages and bedded with LIGNOCEL® bedding material. All mice received phytoestrogen-free diet (C1000 from Altromin, Lage, Germany) and water ad libitum. Experiments included groups of 4-6 mice per cage and were performed at least two times according to institutional and state guidelines. Animal protocols used in this study were approved by the Committee on Animal Welfare of Saxony/Leipzig (Permit Number: TVV01/15).

**Exposure to nBuP and Aza treatment.** To investigate the impact of an intrauterine exposure on weight development in the offspring we exposed pregnant mice to nBuP via subcutaneous (s.c.) injection of 1.75 µg nBuP (Sigma-Aldrich Chemie GmbH, Munich, Germany) in 100 µl corn oil twice per week until weaning when pups were 4 weeks old (perinatal exposure). Control dams received only the vehicle. To investigate a possible involvement of epigenetic alterations offspring were treated i.p. with 160 µg kg$^{-1}$ body weight Aza (Sigma-Aldrich) solved in PBS 3 times per week starting 1 week after birth until weaning[26]. Each exposure protocol was performed at least two times from at least 3 dams (each with 2–5 pups with not more than 3 pups per sex).

**Assessment of weight and metabolic parameters.** Body weight of the pups was measured twice a week and a mean weight per week was calculated for each mouse. At the end of the observation period (12 weeks) whole body composition (fat mass and lean mass) was determined in awake mice based on nuclear magnetic resonance technology using an EchoMRI700™ instrument (Echo Medical Systems, Houston, TX, USA) in the offspring of control and nBuP exposed dams. Food intake was monitored after weaning until week 11. For determination of the adipocyte area visceral adipose tissue was removed and stored into in 4% paraformaldehyde for more than 24 h to complete fixation. Afterwards the tissues were dehydrated, embedded in paraffin and sectioned into 3 µm sections. After a deparaffinization, tissue sections were stained with hematoxylin and eosin (H&E) for the adipocyte area assessment. Insulin tolerance test (ITT) was performed in the offspring 9 weeks after birth. Insulin (0.75 U kg$^{-1}$ body weight) was injected intraperitoneally. For glucose measurements blood of tail vein was taken at four time points at 0, 15, 30 and 60 min after insulin injection. For the glucose tolerance test (GTT) glucose (2 g kg$^{-1}$ body weight) was injected i.p. and the measurement was performed equally. Adiponectin, Leptin, Resistin, Insulin, and acetylated Ghrelin serum concentrations were determined by ELISA using mouse standards according to the manufacturer's guidelines (Mouse Adiponectin, Leptin, Resistin ELISA; R&D Systems/bio-techne, Minneapolis, USA, Mouse/Rat/Human 17-beta Estradiol ELISA; Abcam, Cambridge, UK, Mouse/Rat Insulin ELISA; bertinpharma, Montigny-le-Bretonneux, France and Mouse/Rat acylated Ghrelin ELISA; BioVendor, Karasek, Czech Republic).

**RNA extraction, cDNA synthesis and qPCR.** Dissection of the hypothalamus was conducted from the ventral side of the brain. The optic chiasm was removed away from the anterior portion of the hypothalamus. The mammillary nuclei were dissected from the posterior of the hypothalamus. The entire hypothalamus was prepared including the arcuate, ventromedial, dorsomedial, and paraventricular nuclei. Total RNA was extracted from adipocytes of humans, visceral adipose tissue and hypothalamus of mice by using QIAzol Lysis Reagent (QIAGEN, Hilden, Germany) and RNAeasy Plus Mini Kit (QIAGEN) following manufactures instructions. Two hundred nanogram were used for cDNA synthesis by RevertAid™ H Minus Reverse Transcriptase (Thermo Fisher Scientific, MA, USA). qPCR was performed with BIOTAQ™ DNA polymerase (Bioline GmbH, Luckenwalde, Germany) and SYBRgreen I nucleic acid gel stain (Thermo Fisher Scientific) on a LightCycler 480 (Roche Applied Sciences, Penzberg, Deutschland) with the following cycling programme: 10 min at 95 °C, followed by 45 cycles of 95 °C for 10 s, 20 s at 68 °C and 72 °C for 20 s. All reactions were performed in duplicates. Primers were designed exon spanning. Primer sequences of target genes are listed in Supplementary Table 6. Expression values of target genes were evaluated using 2-ΔΔCT method with U6 small nucleolar RNA (u6), beta-actin (actb), the beta-glucuronidase (gusb, GUSB) and phosphoglycerate kinase 1 (PGK1) as reference genes and normalised to the lowest measured value.

**Pyrosequencing.** For DNA methylation analysis of POMC promoter and enhancers genomic DNA (gDNA) of the hypothalamus was isolated using the DNeasy Blood and Tissue Kit (QIAGEN). Two hundred nanogram were bisulfite converted using the EZ DNA Methylation™ Kit (ZymoResearch, CA, USA). The pyrosequencing assay for DNA-methylation analysis of enhancers was designed on the forward strand and on the reverse complement-strand for POMC promoter using the PyroMark Assay Design version 2.0.1.15 software (Qiagen, POMC enhancer nPE1: forward primer 5'-GTGGGTAAGTTTGAGTTTTGAA TG-3', reverse primer 5´-biotin-ACCCTTCCTCAAAAATACAAAATTC-3', sequencing primer 5'-AGTTTGAGTTTTGAATGT-3', amplicon coordinates: chr12: 3942231-3942372; POMC enhancer nPE2: forward primer 5'-TTTTTT TGTTTTGTGGGGTATGTAGT-3', reverse primer 5´-biotin-AAAAACCCTAA TAAAAAACCCCTTAA-3', sequencing primer: 5'-AATATATGTATTAGTGGA TGAAA-3', amplicon coordinates: chr12: 3944766-3944876; POMC promoter: forward primer 5'-GTTGGGTGGGTGAGTTTT-3', reverse primer 5´-biotin-CAC CATTCTTAATTAAATTCTTCCTAACC-3', sequencing primer: 5'-GTGGGTGA GTTTTGGA-3', amplicon coordinates: chr12:3954708-3954911). The genomic position of the amplicons for DNA-methylation of the POMC promoter and enhancers nPE1/nPE2 were determined based on Rubinstein et al.[33], Drouin et al.[47], and Langlais et al.[48]. (see Supplementary Fig. 7).

Bisulfite treated gDNA was amplified with the HotStar Taq DNA Polymerase Kit (QIAGEN) with the following cycling programme: 15 min at 95 °C, followed by 45 cycles of 94 °C for 30 s, 30 s at 56 °C and 72 °C for 50 s, and a final elongation for 10 min at 72 °C. DNA methylation was assessed by pyrosequencing on a PyroMark Q48 (QIAGEN) following manufacture's instruction.

**Statistical analysis**. LINA study data were evaluated by STATISTICA for Windows, Version 12 (Statsoft Inc., USA) and STATA version 15.1 (StataCorp LLC, USA). A GEE model with exchangeable correlation matrix and cubic splines with knot points at 12, 25, 38.2, 61 and 98,2 months was applied to test for longitudinal differences in BMI between low (1st tertile, reference), medium (2nd tertile) and high (3rd tertile) paraben exposure groups. Logistic regression models were performed to analyse the impact of prenatal paraben exposure on infants' risk for macrosomia at birth and overweight development afterwards (year 2–8). Both models were adjusted for the sex of the child, smoking during pregnancy, parental school education, gestational week at delivery, existence of siblings, breast feeding duration (not for models with only birth weight as outcome) and age of the mother at birth. All confounders were chosen according to their potential weight association after a literature review. To test the equal distribution of parameters in the analysed sub-cohort and the entire LINA cohort the chi squared test was performed. Mann–Whitney-$U$-test was used for the comparison of median paraben levels of mothers that used paraben containing vs. paraben free cosmetic products. Experimental data sets from in vivo and in vitro studies were processed and analysed in GraphPad PRISM 7.02 for windows (GraphPad Software, Inc.) and R version 3.5.1. All $p$-values are derived by GEE models with exchangeable correlation matrix for longitudinal mouse data, ANOVA or unpaired $t$-test.

**Reporting summary**. Further information on research design is available in the Nature Research Reporting Summary linked to this article.

## Data availability
All relevant data supporting the findings of the study are available in this article, its Supplementary Information file, the Source Data file (Figs. 1B–E; 2A–E; 3B–D; 4A–F and Supplementary Figs. 2A–C; 3B–C; 4A–C; 5; 7A–B; 8A–C), or from the corresponding authors on request. However, epidemiological cohort data cannot be provided as an open source file due to ethical declaration issues and can be requested in their analysed version from the corresponding author.

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

## Acknowledgements

We thank Marita Reiprich, Michaela Loschinski, Elena Elter, Susanne Arnold, Beate Fink, Coretta Bauer, and Melanie Bänsch for excellent technical assistance and all participants of the LINA study. This work was supported by the Deutsche Forschungsgemeinschaft (SFB 1052) and the BMBF (IFB AdipositasDiseases Leipzig K7-75).

## Author contributions

K.J., I.L. and T.P. designed and conducted the study and experiments; B.L., K.J. and C.P. analysed the cohort data; B.L., S.S and I.K. performed the cell culture and mouse experiments; B.S., L.S. and T.R. established and performed the paraben analytic; R.S. and B.I.E. performed the reporter gene assays; M.Bo., S.R. and I.L. developed the LINA study design, A.S. assessed the whole body composition; M.Ba. conducted the PCR analysis, U. E.R-K. and M.v.B. prepared the human urine; S.S. and L.T. performed the pyrosequencing, T.P., B.L., K.J., I.L., G.I.S. T.S. and B.I.E. discussed the data with substantial contributions from T.R. and M.v.B.; K.J. and B.L. designed Fig. 5. B.L. and T.P. wrote the paper with substantial inputs from G.I.S., T.S., I.L., KJ, B.I.E., M.v.B. and T.R.

## Competing interests

The authors declare no competing interests.
