## [Peer Review File · Nature Communications]

Reviewers' comments:

Reviewer #1 (Remarks to the Author):

In this paper, Leppert and colleagues examine the effects of developmental exposure to parabens, more specifically butylparaben (nBuP), on early life weight gain. Although parabens are widely used in daily cosmetic and food products, their effects on metabolic regulation, especially when exposure occurs during important periods of development, remain elusive. This translational paper, which includes both human and mice studies, supports the hypothesis that exposure to parabens during perinatal life increases predisposition to obesity in the offspring and likely involves dysregulations of the hypothalamic melanocortin system. Overall, the experiments are well-designed and the paper is well-written and provide novel information. There are, however, several concerns that should be addressed prior to publication:

1. The author's state transgenerational in the introduction and discussion, but they only looked at the first offspring that had direct exposure to nBuP. This is not a transgenerational effect, but rather an effect from a direct developmental exposure. The word transgenerational is misleading the reader to think nBuP has effects on the offspring of the mice born from the dam exposed to nBuP, which would be the F2 generation. This was not tested here. Please revise the language throughout the manuscript accordingly.
2. Do the authors have access to maternal blood/serum during pregnancy? It would be of great interest to see what the difference in paraben levels between the two groups would have been in the blood. The urinary data are highly relevant and great to show that the mouse model is comparable to that of humans, but it does not show what levels a human fetus may be exposed to in utero.
3. Please describe the model portrayed in Figure 2 in greater detail either in the results or discussion. It is unclear what this model details and how it contributes to the manuscript.
4. Metabolic analysis in the human study is superficial and is limited to evaluation of body weights. Addition of data on adiposity and/or food questionnaires would strengthen this paper.
5. A more detailed description of the anatomical landmarks used for dissection of the hypothalamus is needed. Also, because of the hypothalamus is a very heterogeneous brain region that contains a variety of neuronal systems and nuclei playing distinct and opposite role on appetite regulation, analysis of MC4R and Ob-R expression in specific hypothalamic sub-nuclei (versus whole hypothalamus measurements) would provide more useful information.
6. Figure 6 is confusing with the gene expression axes titled "fold change". What is this fold change relative to? It does not look like the control values are set to 1, so what is fold change?
7. It is not clear whether Figure 6 is another cohort of mice than those in Figure 4? Is the increased weight gain and increased food intake effect of nBuP still there? This is not depicted in the figure.
8. Does the administration of Aza also revert the effects perinatal nBuP exposure had on female adipocyte size, the increased leptin serum levels, and the decrease in leptin receptor expression in the hypothalamus?
9. Figure 6E and 6F show interesting results with the use of Aza alone (CON + Aza) where it appears there is actually a decrease in methylation and a decrease in POMC expression. I think this is something the author's should discuss or address since this is the opposite effect one would expect from a methyltransferase inhibitor. It almost appears as if there is some interaction between nBuP and Aza for it to have such a profound effect in the exposed mice.

10. Line 291: The author's describe a citation (36) that details the importance of the enhancer regions for POMC transcriptional regulation but citation 36 is titled "Transplacental passage of antimicrobial paraben preservatives". Please insert the appropriate citation for the statement made in line 288-291.

11. Line 438: The author's state each exposure protocol was performed at least two times from at least 3 dams, but can they specify whether only one and male pup from each litter was used? In developmental exposures like this the litter is the independent unit of measurement since the dam was exposed.

Reviewer #3 (Remarks to the Author):

The work described in the manuscript entitled "Maternal paraben exposure triggers childhood overweight development" addresses the question of whether parabens may be playing an important role in the current obesity trends worldwide. They discuss data obtained from a human cohort of pregnant women and their children, in vitro analyses using stem cells and mouse analysis. They find an association between paraben exposure during in utero development and overweight later in life. However, the results do not prove their conclusions. The sexually dimorphic phenotype is not well addressed using their in vitro model and the comparison of human in vitro data with mouse in vivo data is not necessarily straight forward. Further experiments are required to fill those gaps.

Introduction

There is no information in the introduction about known levels of exposure in humans. This information is present in the discussion but it would be valuable to have it in the introduction to put the subsequent data into context.

There is no justification for the use of those concentrations of parabens in vitro. The concentrations used in the assay are higher than the concentrations that would be expected for humans to be exposed to and they seem higher than those concentrations needed to alter the endocrine system.

Results.

Figure 1: In the figure legend, the description of the panels does not match what the figure shows.

Figure 2: A better description of the figure would be necessary to better interpret it. How were the Z scores calculated? A brief explanation is necessary.

Figure 3: The lipid accumulation calculation should be corrected by the number of cells. What does Cell index mean? IS it a measure of number of cells at each timepoint? In the text there is a call for panel 3E (line 157), but there is not panel E in figure 3. Representative pictures of the oil red o staining should be introduced in the main figure.

Were the human cells male or female cells? This is an important factor considering that the authors find a sexually dimorphic phenotype and should be addressed.

The same adipogenesis assay should be performed using mouse MSCs, which will allow a better discussion on whether the in vivo data observed in mice is consistent with the in vitro data.

Aza analysis. The data generated needs to be taken with caution. Aza demethylates the whole genome. The recovery of the baseline phenotype after treating the animals with AZA is not

necessarily due to a demethylation of the POMC regulatory regions.

Discussion.

Line 223. The authors mentioned they used a transgenerational mouse model which is misleading since it has been accepted that an effect is considered transgenerational if it is seen in the F3 generation after pregnancy exposure of F0 dams.

Line 296-299. The authors discuss that "exposure of adult animals did not show an effect..." but they don't show any adult data.

Reviewer #4 (Remarks to the Author):

This is a well written and novel paper of the association between maternal paraben exposure and childhood overweight development. There are major methodological issues, which need to be addressed.

Major comments:

1. Overweight and Obesity cut-offs: The cut off used to define overweight and obesity is not the standard cut-off proposed by WHO. According to WHO for children under 5 years of age: overweight is weight-for-height greater than 2 standard deviations above WHO Child Growth Standards median; and obesity is weight-for-height greater than 3 standard deviations above the WHO Child Growth Standards median. For children aged between 5-19 years old overweight is BMI-for-age greater than 1 standard deviation above the WHO Growth Reference median; and obesity is greater than 2 standard deviations above the WHO Growth Reference median.
<http://www.who.int/en/news-room/fact-sheets/detail/obesity-and-overweight>
Alternatively, authors could use the International Obesity Task Force (IOTF) revised child BMI age and sex specific cut-offs for overweight and obesity. The cut-offs are available for exact ages by month from 2 to 18 year (<http://www.worldobesity.org/resources/child-obesity/newchildcutoffs/>) (Cole and Lobstein, 2012)
2. Birth weight measures: For birth weight measures, authors should use terms as "large for gestational age neonates" or "macrosomia" (>4000Kg), as it is not recommended to classify neonates as overweight or obese.
3. Questionnaire data: The use of questionnaire data to measure parabens exposure is questionable as the authors already measured metabolites of these chemicals in urine. It is unclear how the use of questionnaire data adds value to the current analysis.
4. Mediation analysis: Mediation analysis by cosmetic use does not follow the classical definition of a mediator, ie that the independent variable influences the (non-observable) mediator variable, which in turn influences the dependent variable. In other words, cosmetic use and paraben exposure should be both considered as independent variables in the causal pathway linking maternal paraben exposure and birth weight.
5. It would be helpful to include a diagram describing the translational research design of the study for example how findings from the population human study could be used to inform the in vivo and in vitro studies and the link between these different research designs.

Answers to the reviewers' comments on the manuscript:

"Maternal paraben exposure triggers childhood overweight development"

We thank the reviewers and the editorial team for the extremely helpful comments regarding our manuscript "Maternal paraben exposure triggers childhood overweight development".

We addressed all issues raised by the reviewers in the text below and performed several additional experiments. The results of these experiments as well as further additional data are now included in the revised manuscript and described in the text. We very much hope that we could clarify the open questions and thank you for taking the time to review our manuscript.

Reviewer #1 (Remarks to the Author):

In this paper, Leppert and colleagues examine the effects of developmental exposure to parabens, more specifically butylparaben (nBuP), on early life weight gain. Although parabens are widely used in daily cosmetic and food products, their effects on metabolic regulation, especially when exposure occurs during important periods of development, remain elusive. This translational paper, which includes both human and mice studies, supports the hypothesis that exposure to parabens during perinatal life increases predisposition to obesity in the offspring and likely involves dysregulations of the hypothalamic melanocortin system. Overall, the experiments are well-designed and the paper is well-written and provide novel information. There are, however, several concerns that should be addressed prior to publication:

COMMENT:

The author's state transgenerational in the introduction and discussion, but they only looked at the first offspring that had direct exposure to nBuP. This is not a transgenerational effect, but rather an effect from a direct developmental exposure. The word transgenerational is misleading the reader to think nBuP has effects on the offspring of the mice born from the dam exposed to nBuP, which would be the F2 generation. This was not tested here. Please revise the language throughout the manuscript accordingly.

Response:

The wording was corrected accordingly throughout the manuscript.

COMMENT:

Do the authors have access to maternal blood/serum during pregnancy? It would be of great interest to see what the difference in paraben levels between the two groups would have been in the blood. The urinary data are highly relevant and great to show that the mouse

model is comparable to that of humans, but it does not show what levels a human fetus may be exposed to in utero.

Response:

We thank the reviewer for bringing up this important issue we haven't addressed appropriate in the first version of our manuscript.

Recent studies have shown that for paraben exposure assessment urine measurements are the method of choice. For endocrine disruption chemicals in general it was shown that the biomarker concentration in urine is representative of an on-going exposure during a specific period of time (Teitelbaum et al. 2008, for phthalates). They also reported for paired paraben samples that only a linear relationship between the oral dose and urine concentration was detected whereas for oral dose and serum concentration it was not (Teitelbaum et al. 2016). Furthermore, analyses in the urine also have some technical advantages: Urine concentrations of total parabens are usually higher than in serum samples (up to 970 times more for methyl paraben and up to 650 times for propyl paraben) as shown by Hines et al. (2015) who measured paraben concentrations in serum and urine samples of mothers. As the limit of detection is comparable in the two matrices, it is more likely to detect representative paraben concentrations in urine, rather than serum. Due to the fast metabolization of parabens in the liver (half-life in humans less than 24 h) and direct urinary clearance of conjugated metabolites, it's reasonable to detect conjugated metabolites in urine rather than serum (Campbell et al. 2015).

We could fully agree with the reviewer that it would be an interesting addition to know the exact paraben levels also in maternal blood to get a closer idea of the potential exposure concentration of the fetus via the placenta. Within the LINA study we were not able to do these additional measurements since we have only limited access to maternal serum from the pregnancy period anymore. However, data from earlier published studies provide information on the urine / serum ratio of parabens including butylparaben (Campbell et al. 2015). Thus, knowing the exact urine concentration the corresponding blood concentration can be estimated. To make this point more clear and provide an idea on fetal exposure concentrations in our study we have added the following information to the manuscript (line 79):

Paraben level vary among the different parabens, tissue and time after topical application with previously reported concentration of 0.5 µg/l butyl paraben to 43.9 µg/l methyl paraben in urine samples and about 4-times lower serum level¹⁸. The fast metabolization of parabens in the liver (half-life in humans less than 24 h) and direct urinary clearance of conjugated metabolites may explain the higher urine concentration compared to serum. Thus, as the limit of detection is comparable in the two matrices, it is more likely to detect representative paraben concentrations in urine, rather than in serum.

COMMENT:

Please describe the model portrayed in Figure 2 in greater detail either in the results or discussion. It is unclear what this model details and how it contributes to the manuscript.

Response:

According to the critical comments from all reviewers about the mediation model shown in Figure 2 we decided to take these results out of the manuscript. To better explain that

cosmetic products are a possible source of paraben exposure we added a paragraph to the introduction about paraben absorption via the skin and paraben exposure in other cohorts (line 75). We also added the information regarding the potential source of parabens drawn from our present analyses (application via cosmetic products) in our summary scheme (which was suggested by reviewer 4).

Parabens are alkyl esters of p-hydroxybenzoic acid (PHBA) with antimicrobial and antifungal properties frequently used as preservatives in cosmetic products, toiletries, food (E214-219), and pharmaceuticals. They are found in the majority of leave-on cosmetics and in rinse-off products¹⁶. Parabens enter the human body mainly through ingestion or skin absorption and can commonly be detected in urine, blood and breast milk.

COMMENT:

Metabolic analysis in the human study is superficial and is limited to evaluation of body weights. Addition of data on adiposity and/or food questionnaires would strengthen this paper.

Response:

We agree with the reviewer that additional data on adiposity or food questionnaires would strengthen our analysis. Unfortunately, we do not have qualitative food frequency questionnaires available in LINA at the time points of interest shown in the present manuscript to gather that information. However, we were able to include Bioelectrical Impedance Analyses (BIA) at the age of 8 years to support our overweight classification. We could show that children classified as overweight at 8 years of age had over 60% higher body fat mass percentage at the same age compared to non-overweight children (fat mass overweight = 19.3%, n=58; fat mass normal weight = 14.9%, n=138, p<0.001). Unfortunately, the number of cases with BIA measurements at the age of eight years and maternal urine paraben concentration measurements is too small to assess an association between BIA data and paraben exposure.

We added the information on validation of obesity assessment via BIA data into the Methods section (line 394).

We were able to verify the OW classification based on a Bioelectrical Impedance Analysis (BIA) performed at children's age of eight. Children classified as OW had about 30 % higher body fat mass percentage at 8 years of age compared to normal weight children (fat mass overweight = 19.3%, n=58; fat mass normal weight = 14.9%, n=138, p<0.001).

COMMENT:

A more detailed description of the anatomical landmarks used for dissection of the hypothalamus is needed. Also, because of the hypothalamus is a very heterogeneous brain region that contains a variety of neuronal systems and nuclei playing distinct and opposite role on appetite regulation, analysis of MC4R and Ob-R expression in specific hypothalamic sub-nuclei (versus whole hypothalamus measurements) would provide more useful information.

Response:

As suggested by the reviewer we have now included a more detailed description for dissection of the hypothalamus in the Methods section (line 503):

„Dissection of the hypothalamus was conducted from the ventral side of the brain. The optic chiasm was removed away from the anterior portion of the hypothalamus. The mammillary nuclei were dissected from the posterior of the hypothalamus. The entire hypothalamus was prepared including the arcuate, ventromedial, dorsomedial, and paraventricular nuclei.“

We agree with the reviewer that the dissection of hypothalamic sub-nuclei would be a more sophisticated method and could improve the specificity of MC4R and Ob-R expression analysis. However, the significantly altered POMC expression observed in the present analyses for the offspring from nBuP-exposed dams compared to the control animals is valid as an explanation for the neuronal appetite dysregulation when analyzed in the whole hypothalamus. POMC expression is mainly located in the nucleus arcuatus (compared to other hypothalamic nuclei) and our shown results for POMC expression are therefore not to be seen as the mean of different hypothalamic area but rather represent the expression change based on alterations in the ARC. Due to the need of a substantial number of mice that would be necessary to gain qualitative data for gene expression after dissection of the hypothalamus into sub-nuclei we decided not to repeat the whole experiment (maternal exposure, offspring weight development etc.). In our opinion the gain in information from that additional experiment does not legitimate all efforts and most importantly relevant issues on animal welfare.

COMMENT:

Figure 6 is confusing with the gene expression axes titled “fold change”. What is this fold change relative to? It does not look like the control values are set to 1, so what is fold change?

Response:

Thank you for this comment. The labelling of the axes with “Fold change” was not correct. The figure shows expression values of target genes evaluated using $2^{-\Delta\Delta CT}$ method with several reference genes and normalized to the lowest measured value. We changed the labelling of the axis in Figure 5 (former Figure 6) and Supplementary Figure S9, accordingly.

COMMENT:

It is not clear whether Figure 6 is another cohort of mice than those in Figure 4? Is the increased weight gain and increased food intake effect of nBuP still there? This is not depicted in the figure.

Response:

The mice in Figure 5 (former Figure 6) are the same cohort than in Figure 3 (former Figure 4). To make the increased weight gain and increased food intake more apparent, we additionally mark the significant differences between the control animals and the offspring from nBuP-exposed dams in Figure 5 (former Figure 6).

COMMENT:

Does the administration of Aza also revert the effects perinatal nBuP exposure had on female adipocyte size, the increased leptin serum levels, and the decrease in leptin receptor expression in the hypothalamus?

Response:

As suggested by the reviewer we added further data regarding adipocyte size, leptin serum levels, and the leptin receptor expression in the hypothalamus in response to the administration of Aza as new Supplementary Figure S9 into the manuscript.

COMMENT:

Figure 6E and 6F show interesting results with the use of Aza alone (CON + Aza) where it appears there is actually a decrease in methylation and a decrease in POMC expression. I think this is something the author's should discuss or address since this is the opposite effect one would expect from a methyltransferase inhibitor. It almost appears as if there is some interaction between nBuP and Aza for it to have such a profound effect in the exposed mice.

Response:

The use of Aza alone (CON + Aza) neither showed significant results nor a trend (all $p > 0.1$). We never consider such data as relevant results and would avoid a speculative discussion. However, to address the needful caution in interpreting the data obtained from Aza treatment we included a statement in the discussion (line 305).

Results of Aza treatment experiments have always to be interpreted with caution since the Aza treatment affects the entire genome. However, comparing the Aza effect in controls and nBuP exposed animals there is some evidence for an involvement of DNA hypermethylation in phenotype development. Anyhow, further investigations are needed to clarify the epigenetic alterations more specifically. We also cannot exclude that additional pathways might be involved in mediating the nBuP-induced increased weight.

COMMENT:

Line 291: The author's describe a citation (36) that details the importance of the enhancer regions for POMC transcriptional regulation but citation 36 is titled "Transplacental passage of antimicrobial paraben preservatives". Please insert the appropriate citation for the statement made in line 288-291.

Response:

Thank you for this comment. The citation was not correct. We now inserted the appropriate citation for the statement made in the discussion.

COMMENT:

Line 438: The author's state each exposure protocol was performed at least two times from at least 3 dams, but can they specify whether only one and male pup from each litter was used? In developmental exposures like this the litter is the independent unit of measurement since the dam was exposed.

Response:

As suggested by the reviewer we specified the number of pups used per exposed dam in the Methods section. We used 2 to 5 pups from the dams but never more than 3 pups from the same sex. Therefore, the group composition was at least 2-3 pups/sex from 3 dams. We are aware that the litter is the independent unit in regard to the exposure. However, for our analysis we handled the littermates as individuals to consider possible individual differences in constitution as well as to keep the number of animals within reasonable limits.

Reviewer #3 (Remarks to the Author):

The work described in the manuscript entitled ‘Maternal paraben exposure triggers childhood overweight development’ addresses the question of whether parabens may be playing an important role in the current obesity trends worldwide. They discuss data obtained from a human cohort of pregnant women and their children, in vitro analyses using stem cells and mouse analysis. They find an association between paraben exposure during in utero development and overweight later in life. However, the results do not prove their conclusions. The sexually dimorphic phenotype is not well addressed using their in vitro model and the comparison of human in vitro data with mouse in vivo data is not necessarily straight forward. Further experiments are required to fill those gaps.

Introduction

COMMENT:

There is no information in the introduction about known levels of exposure in humans. This information is present in the discussion but it would be valuable to have it in the introduction to put the subsequent data into context.

Response:

We have now added information about known paraben exposure, distribution and absorption (if known) to the introduction and discussion.

For example line 79: Parabens are alkyl esters of p-hydroxybenzoic acid (PHBA) with antimicrobial and antifungal properties frequently used as preservatives in cosmetic products, toiletries, food (E214-219), and pharmaceuticals. They are found in the majority of leave-on cosmetics and in rinse-off products¹⁶. Parabens enter the human body mainly through ingestion or skin absorption and can commonly be detected in urine, blood and breast milk^{17, 18}. Paraben level vary among the different parabens, tissue and time after topical application with previously reported concentration of 0.5 µg/l butyl paraben to 43.9 µg/l methyl paraben in urine samples and about 4-times lower serum level¹⁸.

COMMENT:

There is no justification for the use of those concentrations of parabens in vitro. The concentrations used in the assay are higher than the concentrations that would be expected for humans to be exposed to and they seem higher than those concentrations needed to alter the endocrine system.

Response:

We chose paraben concentrations for our cell culture experiments based on previous publications [Hu et al.] and the assumption that 0.5 µM (the lowest concentration used) are approximately equivalent to the highest paraben concentrations found in LINA for MeP.

To address the reviewers' comment we have included an additional paragraph into the discussion (line 268):

„To identify the underlying mechanisms of the nBuP-induced weight gain we first investigated a possible effect on adipocyte differentiation using a human mesenchymal stem cell assay and a murine pre-adipocyte culture model. Here, we chose paraben concentrations based on previous publications [Hu et al.] and the assumption that 0.5 μ M (the lowest concentration used) are approximately equivalent to the highest paraben concentrations found in LINA for MeP. „

Results

COMMENT:

Figure 1: In the figure legend, the description of the panels does not match what the figure shows.

Response:

Thank you for this comment. We corrected the figure legend, accordingly.

COMMENT:

Figure 2: A better description of the figure would be necessary to better interpret it. How were the Z scores calculated? A brief explanation is necessary.

Response:

We decided to take the mediation model shown in Figure 2 out of the manuscript. We agree with the reviewers that the mediation model does not add any substantial information to the study that goes beyond the other analysis.

According to the usage of the z-scores we have now recalculated our data since reviewer 4 had also raised concerns on the underlying calculation/classification. Therefore, we have reclassified our complete data according to the detailed monthly references from the International Obesity task force/World obesity forum from age 2-8 years (without calculation of z-scores but rather using age and gender specific infant BMI cut-off levels to classify overweight in our LINA cohort).

COMMENT:

Figure 3: The lipid accumulation calculation should be corrected by the number of cells. What does Cell index mean? IS it a measure of number of cells at each timepoint? In the text there is a call for panel 3E (line 157), but there is not panel E in figure 3. Representative pictures of the oil red o staining should be introduced in the main figure.

Response:

Oil Red O staining was assessed via absorption at 510 nm with an ELISA reader. Hence, staining was assessed within the entire well to reduce bias. However, we do agree that changes of Oil Red O staining could occur due to increased fatty acid accumulation or cell proliferation and we would not have been able to distinguish between both. However, cell toxicity assays (shown in the supplementary analysis) did not indicate an effect of paraben exposure on the total cell count after differentiation.

Furthermore, as suggested by the reviewer we added an appropriate description of the impedance-based life cell imaging system and have now included that in the methods section (line 412):

“Proliferation and differentiation were monitored in real-time by the impedance-based xCELLigence MP System (ACEA Biosciences Inc., San Diego, USA) on a microelectrode 96 well E-View-Plate (ACEA Biosciences Inc.). A cell index was assessed every 10 min by normalizing an electrical impedance measurement to a bland value for each well. Measurements were paused for media changes.”

We corrected the Figure references accordingly.

Finally, we have additionally included representative pictures of Oil Red O stained cells in Figure 2 (former Figure 3) to further strengthen our conclusion that there are only marginal differences in adipocyte cell differentiation due to nBuP exposure.

COMMENT:

Were the human cells male or female cells? This is an important factor considering that the authors find a sexually dimorphic phenotype and should be addressed.

Response:

Cells ordered from ATCC which were used in the present analyses were derived from females. We added this information in the Methods section. Based on our project results we were aware of that aspect and tried to order specifically male adipocyte derived mesenchymal stem cell to additionally perform our parabens exposure in male cells. However, ATCC was not able to supply male donors at that time. Therefore, we were not able to verify our gender hypothesis on in-vitro basis.

COMMENT:

The same adipogenesis assay should be performed using mouse MSCs, which will allow a better discussion on whether the in vivo data observed in mice is consistent with the in vitro data.

Response:

We agree with the reviewer that an in vitro analysis in a murine model regarding the nBuP effects would be a valuable addition within our translational study design. Therefore, we performed similar analyses shown for the human MSCs in a validated mouse pre-adipocyte culture model (3T3L1, Ruiz-Ojeda et al., 2016). Comparable to the human in vitro analyses we could not detect a direct effect of nBuP on adipogenesis up to a concentration of 10 μ M (see additional Figure S4 in the Supplement). The findings were mentioned in the Results section and included in the new summary Figure 6 describing the translational research design.

COMMENT:

Aza analysis. The data generated needs to be taken with caution. Aza demethylates the whole genome. The recovery of the baseline phenotype after treating the animals with AZA is not necessarily due to a demethylation of the POMC regulatory regions.

Response:

We agree with the reviewer that the data obtained by the treatment with Aza need to be taken with caution. Therefore, we included a statement in the discussion to address this issue (line 305).

Results of Aza treatment experiments have always to be interpreted with caution since the Aza treatment affects the entire genome. However, comparing the Aza effect in controls and nBuP exposed animals there is some evidence for an involvement of DNA hypermethylation in phenotype development. Anyhow, further investigations are needed to clarify the epigenetic alterations more specifically. We also cannot exclude that additional pathways might be involved in mediating the nBuP-induced increased weight.

Discussion

COMMENT:

Line 223. The authors mentioned they used a transgenerational mouse model which is misleading since it has been accepted that an effect is considered transgenerational if it is seen in the F3 generation after pregnancy exposure of F0 dams.

Response:

The wording was corrected accordingly throughout the manuscript.

COMMENT:

Line 296-299. The authors discuss that “exposure of adult animals did not show an effect...” but they don’t show any adult data.

Response:

In the supplementary Figure S5 (formerly S3) we show the effect of nBuP exposure on body weight, food intake and leptin levels in adult mice. The data were also mentioned in the results section.

Reviewer #4 (Remarks to the Author):

This is a well written and novel paper of the association between maternal paraben exposure and childhood overweight development. There are major methodological issues, which need to be addressed.

COMMENT:

Overweight and Obesity cut-offs: The cut off used to define overweight and obesity is not the standard cut-off proposed by WHO. According to WHO for children under 5 years of age: overweight is weight-for-height greater than 2 standard deviations above WHO Child Growth Standards median; and obesity is weight-for-height greater than 3 standard deviations above the WHO Child Growth Standards median. For children aged between 5-19 years old overweight is BMI-for-age greater than 1 standard deviation above the WHO Growth Reference median; and obesity is greater than 2 standard deviations above the WHO Growth Reference median. <http://www.who.int/en/news-room/fact-sheets/detail/obesity-and-overweight>. Alternatively, authors could use the International Obesity Task Force (IOTF) revised child BMI age and sex specific cut-offs for overweight and obesity. The cut-offs are

available for exact ages by month from 2 to 18 year (<http://www.worldobesity.org/resources/child-obesity/newchildcutoffs/>) (Cole and Lobstein, 2012).

Response:

We thank the reviewer for this valuable comment. As suggested above, we have reclassified our complete data according to the detailed monthly references from the International Obesity task force/World obesity forum from age 2-8 years. Since herein no data are available for children younger than 2 years we have classified “birth weight” for “large for gestational age (LGA)” or “not LGA” (as mentioned in comment 2 below). Furthermore, since our initial submission, we are able to add additional weight and height data for year 8 from the regular school based U-examinations (retrospectively assessed via questionnaire) which resulted in additional 22 cases. To further enhance data transparency we have also included a longitudinal overview about the raw BMI development over the years when compared between high vs. low paraben exposure (Figure 1 as well as Supplementary Figure S2). In summary, we were still able to show our main results indicating that prenatal nBuP and iBuP exposure enhances the risk to become overweight in infancy (year 2-8). We were not able to show effects of prenatal BuP exposure on birth weight after the suggested classification into LGA or Non-LGA.

The manuscript part focusing on the association between prenatal paraben exposure and body weight development has been revised as follows (line 127):

Further we investigated the impact of prenatal paraben exposure on children’s weight development up to the age of 8 years. By applying an ANOVA analysis on raw longitudinal BMI-data we found significant differences between prenatally low (1st tertile) and high (3rd tertile) exposed children for iBuP and nBuP (Figure 1) but not for MeP, EtP, and PrP (Supplementary Figure S2). Thereby, the BuP related BMI increase was more evident in girls compared to boys. Next, we defined overweight children using age and sex specific cut-offs from the International Obesity Task Force (IOTF). Adjusted logistic regression models were applied to study a potential risk increase resulting from prenatal paraben exposure comparing overweight and non-overweight children at birth (considering large for gestational age as “overweight”) and afterwards (Table 2). Since reference data are not available for one year-old children, only the time window between age two and eight could be considered for this analysis. While high exposure to parabens did not affect the risk for overweight at birth, there was evidence for the long chain parabens iBuP and nBuP to increase the risk for overweight in later infancy (Table 2 for BuP, Supplementary Table S3 for MeP, EtP and PrP). Indeed, there was some evidence for a stronger effect of prenatal butyl paraben exposure on overweight risk at later infancy in girls compared to boys without reaching significance level in this regression analysis due to limitations in sample size after gender stratification.

COMMENT:

Birth weight measures: For birth weight measures, authors should use terms as “large for gestational age neonates” or “macrosomia” (>4000Kg), as it is not recommended to classify neonates as overweight or obese.

Response:

As suggested by the reviewer, we have reclassified our newborn children/weight at birth according to this cutoff and term. We are now using the definition “large for gestational

age (LGA)” for all children with birthweight >4000g (with is almost identical to the 90th percentile within our LINA population (4030g) compared to “not LGA”. Furthermore, we have reclassified our complete data according to the detailed monthly references from the International Obesity task force/World obesity forum from age 2-8 years.

COMMENT:

Questionnaire data: The use of questionnaire data to measure parabens exposure is questionable as the authors already measured metabolites of these chemicals in urine. It is unclear how the use of questionnaire data adds value to the current analysis.

Response:

We thank the reviewer for addressing this important point. Obviously, the description of our results was not clear enough to explain our strategy. We didn't use questionnaire data to assess paraben exposure— this was done by measured paraben concentrations in the maternal urine (see Methods part: Analysis of urinary paraben concentrations in human samples). We rather used the questionnaire data to document a potentially relevant source of parabens to highlight potential action points for policy makers (see Methods part: Assessment of cosmetic product application). To make this clearer we revised the results part as follows (line 107):

Maternal paraben exposure was assessed by urine measurements showing higher exposure level for methyl paraben and lower for butyl parabens (Supplementary Table S2). As a potential source of paraben exposure the usage of cosmetic products during pregnancy was assessed by questionnaires. Indicated cosmetic products were searched for their paraben content with the TOXFOX app (described in Methods) and categorized in leave-on and rinse-off products.

COMMENT:

Mediation analysis: Mediation analysis by cosmetic use does not follow the classical definition of a mediator, ie that the independent variable influences the (non-observable) mediator variable, which in turn influences the dependent variable. In other words, cosmetic use and paraben exposure should be both considered as independent variables in the causal pathway linking maternal paraben exposure and birth weight.

Response:

Thank you for this comment. We decided to take the mediation model shown in former Figure 2 out of the manuscript. We agree with the reviewers that the mediation model does not add any substantial information to the study that goes beyond the other analysis. We do agree that this is not a classical case of a mediation analysis. However, we were able to show that cosmetic use and urinary paraben concentrations were not independent (Table 1) and cosmetic use can be considered as one of the sources of paraben exposure. There may still be other ingredients in cosmetics that might cause childhood overweight (phthalates, BPA, etc.) and we have now included a point in the discussion on that.

COMMENT:

It would be helpful to include a diagram describing the translational research design of the

study for example how findings from the population human study could be used to inform the in vivo and in vitro studies and the link between these different research designs.

Response:

We have provided a summary scheme (Figure 6) describing the translational research design reported in the present manuscript. The scheme is showing that the findings from the prospective birth cohort study LINA were used to establish hypothesis driven approaches in human and mouse in vitro studies. Based on these results overall mechanistic analyses were addressed in the final mouse in vivo settings.

We thank again the reviewers for their valuable comments. Please, do not hesitate to contact us if there are still points to be clarified.

Thank you for taking the time to review our manuscript.

Sincerely yours,

Tobias Polte, Beate Leppert, Kristin Junge, Irina Lehmann

Reviewers' comments:

Reviewer #1 (Remarks to the Author):

The revised manuscript by Leppert and colleagues is improved and the responses/revisions related to my comments are satisfactory. I have no further comments or concerns.

Reviewer #3 (Remarks to the Author):

The authors described the effect of prenatal exposure to parabens and its association with body weight and adiposity. Although the analysis performed in human samples are innovative and significant, there are some concerns previously raised that have not been addressed and that are important prior to publication in this journal.

Major concerns:

Sexual dimorphism: The sexually dimorphic phenotype was not addressed. There are commercially available human MSCs from both genders (e.g. Promocell).

Adipogenesis assay: The authors show that there is no difference in lipid accumulation in their adipogenesis assays in both human MSCs and 3T3-L1 cells. However, there are some technical inconsistencies.

In both cases, there is not a positive control that induces lipid accumulation to determine whether the assay is working or not. Therefore, the results are not conclusive. One such positive control group could be cells that were exposed to rosiglitazone, which is a potent PPAR γ activator. One of the inconsistencies comes from the fact that the adipogenic cocktail for 3T3-L1 cells already has rosiglitazone. Since rosiglitazone is a potent inducer of adipogenesis, it may be preventing the detection of any potential induction by the parabens.

Additionally, 3T3-L1 cells do not substitute the analysis of mouse MSCs. 3T3-L1 cells are already preadipocytes, whereas MSCs are uncommitted precursors that have not yet made the decision about the differentiation pathways. Therefore, the comparison between human MSCs and 3T3-L1 is not straight forward. There are commercially available mouse MSCs (e.g. from Cygen)

In summary,

- the authors should carry a positive adipogenic control to demonstrate that their assay is indeed working.
- The use of 3T3-L1 does not satisfy the previous concern about checking the effect in other species.
- The sexually dimorphic response to parabens have yet to be addressed.

Statistical analyses: In longitudinal analyses such as those represented in figures 3A-C and 5D, it is more appropriate to use two-way ANOVA to account for the changes that occurred throughout the whole experiment.

Minor comments

Figure 4C. PPAR gamma is mislabeled as PPAR γ

Reviewer #4 (Remarks to the Author):

While the authors did adequately respond to most of the reviewers' comments, the analysis of prenatal paraben exposure and child BMI growth seem to be still a stretch.

1. The use of the term "large for gestational age" is not correct. LGA neonates are defined as live-born infants above the 90th percentile of birth weight for gestational age in a referent population. If the authors use the definition of birth weight ≥ 4000 g without taking into account percentiles, then, they should use the term "macrosomia" and adjust their models for gestational age. The term "overweight/obese" should not be used for birth weight measures.

2. The authors define infancy as years 2-8. This is not correct as the age range corresponds to early-to mid-childhood. Infancy according to CDC is defined as the age period from birth-1 year of age.

3. If the authors want to use a categorical outcome for rapid growth in infancy they could use the term "Rapid growth" defined as a Z score change of greater than 0.67 SD between birth and 6 or 12 months of age, and slow/average growth will be defined as a Z score change of equal or below 0.67 SD (Mendez et al. 2011; Monteiro and Victora 2005).

4. The use of two-way ANOVA for the longitudinal analysis on prenatal paraben exposure and child BMI is not appropriate as it does not account for covariates and repeated outcomes. The authors should assess associations of prenatal exposures with longitudinal changes in BMI, using linear mixed effects models or similar techniques for the longitudinal repeated outcome measures. Potentially nonlinear functional forms of such associations should be examined by using flexible modeling techniques (e.g., splines).

Answers to the reviewers' comments on the manuscript:

"Maternal paraben exposure triggers childhood overweight development"

We thank the reviewers and the editorial team for the helpful comments regarding our manuscript "Maternal paraben exposure triggers childhood overweight development".

We addressed all issues raised by the reviewers in the text below and performed additional experiments. The results of these experiments are now included in the revised manuscript and described in the text. We very much hope that we could clarify the open questions and thank you for taking the time to review our manuscript.

Reviewer #1 (Remarks to the Author):

The revised manuscript by Leppert and colleagues is improved and the responses/revisions related to my comments are satisfactory. I have no further comments or concerns.

Reviewer #3 (Remarks to the Author):

The authors described the effect of prenatal exposure to parabens and its association with body weight and adiposity. Although the analysis performed in human samples are innovative and significant, there are some concerns previously raised that have not been addressed and that are important prior to publication in this journal.

COMMENT:

Sexual dimorphism: The sexually dimorphic phenotype was not addressed. There are commercially available human MSCs from both genders (e.g. Promocell).

Response:

We agree with the reviewer that the adipocyte differentiation assay using MSCs from human female and male donors would make the study more complete. Therefore, we have contacted Promocell and ATCC again. However, both companies have informed us that they are still unable to provide MSCs from a male donor. [Redacted]

However, the sexually dimorphic response to parabens has been addressed in our manuscript both in the human study as well as in the animal experiment. In particular to address a potential sexually dimorphic response in vivo data are much more meaningful than in vitro data using isolated cell populations without the body environment.

COMMENT:

Adipogenesis assay: The authors show that there is no difference in lipid accumulation in their adipogenesis assays in both human MSCs and 3T3-L1 cells. However, there are some technical inconsistencies. In both cases, there is not a positive control that induces lipid accumulation to determine whether the assay is working or not. Therefore, the results are not conclusive. One such positive control group could be cells that were exposed to rosiglitazone, which is a potent PPAR γ activator. One of the inconsistencies comes from the fact that the adipogenic cocktail for 3T3-L1 cells already has rosiglitazone. Since rosiglitazone is a potent inducer of adipogenesis, it may be preventing the detection of any potential induction by the parabens.

Response:

We agree with the reviewer that we have to demonstrate the functioning of our adipocyte differentiation assays. According to the reviewer's suggestion in the next comment we established a murine adipocyte derived mesenchymal stem cell (MSC) assay to replace the 3T3L1 model we have presented in an earlier version of the manuscript. To consider your comment rosiglitazone provided by Cyagen was only used as positive control (see Supplementary Figure S4 and the figure below with additional adipocyte-related genes). For the human MSCs, we would like to refer to our already published data (Junge et al., Clin Epigenetics, 2018) with another chemical (Bisphenol A) using this human adipocyte differentiation assay. Experiments for both chemicals have been done in parallel. With Bisphenol A we could show a positive effect on adipocyte differentiation (see Figure 5 in Junge et al. 2018). Our Bisphenol A data clearly demonstrate the functioning of the human adipocyte differentiation assay. We added an additional sentence into the discussion to point this out (page 13, line 276).

Effect of nBuP exposure on murine adipocyte differentiation.

In vitro adipocyte differentiation from murine mesenchymal stem cells (MSC) in the presence of nBuP and rosiglitazone (POS). (A) Representative Oil Red O stained pictures after differentiation (B) Triglyceride storage of adipocytes assessed via Oil Red O staining. (C) Gene expression of leptin (LEP), adiponectin (ADIPOQ), transcription factor peroxisome proliferator-activated receptor gamma (PPAR γ), lipoprotein lipase (LPL), fatty acid binding protein 4 (FAB4) and CCAAT/enhancer-binding protein (CEBP α). Data are expressed as mean \pm SEM of n = 3 experiments.

COMMENT:

Additionally, 3T3-L1 cells do not substitute the analysis of mouse MSCs. 3T3-L1 cells are already preadipocytes, whereas MSCs are uncommitted precursors that have not yet made the decision about the differentiation pathways. Therefore, the comparison between human MSCs and 3T3-L1 is not straight forward. There are commercially available mouse MSCs (e.g. from Cygen)

Response:

As suggested by the reviewer we performed additional experiments to analyze the effect of nBuP on adipocyte differentiation using murine MSCs. As shown in the revised Supplementary Figure S4 nBuP exposure had no effect on adipocyte differentiation compared to the ethanol control while rosiglitazone increased the amount of triglyceride storage and induced changes in adipogenic gene expression profile. The 3T3L1 data were now excluded.

COMMENT:

Statistical analyses: In longitudinal analyses such as those represented in figures 3A-C and 5D, it is more appropriate to use two-way ANOVA to account for the changes that occurred throughout the whole experiment.

Response:

We agree with the reviewer that another statistical approach might be more appropriate for longitudinal data. Longitudinal analyses have now been carried out applying GEE models with unstructured correlation matrix to account for repeated measurements that are not independent of each other. GEE models were chosen to be consistent with longitudinal analysis of human data (see reviewer 4) with respect to paraben exposure and weight development, where the opportunity to adjust for confounders is crucial. Please also see the last comment/response below from Reviewer 4.

GEE results are consistent with test results presented earlier showing a statistically significant higher body weight and food intake of female mice in comparison to controls and no evidence for a paraben effect on body weight/food intake in male mice. In addition we were also able to show the reducing effect by AZA treatment. We have adapted the methods (statistical analyses) and results part/figures accordingly.

Additionally, as suggested by the reviewer we have also performed a two-way ANOVA analysis as shown below. The test provided similar results as the GEE analysis but effects may be inflated due to the high correlation between repeated measurements. Therefore, as stated above we decided to include the GEE model in the manuscript.

ANOVA analysis

Figure 2A. Two-way ANOVA comparing bodyweight per week of controls and prenatally with nBuP exposed offspring over the observation period (interaction nBuP:week)

Females	F value	P value	Males	F value	P value
con vs. nBuP	39.55	4.72e-07 ***	con vs. nBuP	0.623	0.435
nBuP:week	0.253	0.615	nBuP:week	0.241	0.624

Figure 2C. Two-way ANOVA comparing food intake per week of controls and prenatally with nBuP exposed offspring over the observation period (interaction nBuP:week)

Females	F value	P value	males	F value	P value
con vs. nBuP	13.21	0.003 **	con vs. nBuP	0.21	0.65
nBuP:week	6.30	0.014 *	nBuP:week	1.86	0.17

New Figure 4D. Two-way ANOVA comparing bodyweight per week of controls, prenatally with AZA exposed controls, nBuP exposed and nBuP+AZA exposed offspring over the observation period (interaction with week)

Weight	F value	P value				
Status	56.27	<2x10-16				
status:week	3.18	0.024				
Pairwise comparison (Tukey post-hoc test)						
			Diff	Lower CI	Upper CI	p-value
Con_AZA-Con			1.16	0.48	1.85	<2x10-16
BuP_Con			1.24	0.64	1.84	<2x10-16
BuP_AZA-BuP			1.54	0.93	2.15	<2x10-16

Figure 4D. Two-way ANOVA comparing food intake per week of controls, prenatally with AZA exposed controls, nBuP exposed and nBuP+AZA exposed offspring over the observation period (interaction with week)

Food intake	F value	P value				
Status	9.41	6.5x10-6				
status:week	5.51	0.001				
Pairwise comparison (Tukey post-hoc test)						
			Diff	Lower CI	Upper CI	p-value
Con_AZA-Con			0.70	-0.14	1.54	0.138
BuP_Con			1.90	0.96	2.83	0.000
BuP_AZA_Con			0.97	0.11	1.83	0.021
BuP-Con_AZA			1.20	0.26	2.14	0.06
BuP_AZA-Con_AZA			0.27	-0.59	1.13	0.851
BuP_AZA-BuP			-0.93	-1.89	0.03	0.061

COMMENT:

Figure 4C. PPAR gamma is mislabeled as PPARy

Response:

We corrected the labeling.

Reviewer #4 (Remarks to the Author):

While the authors did adequately respond to most of the reviewers' comments, the analysis of prenatal paraben exposure and child BMI growth seem to be still a stretch.

COMMENT:

The use of the term "large for gestational age" is not correct. LGA neonates are defined as live-born infants above the 90th percentile of birth weight for gestational age in a referent population. If the authors use the definition of birth weight 4000g without taking into account percentiles, then, they should use the term "macrosomia" and adjust their models for gestational age. The term "overweight/obese" should not be used for birth weight measures.

Response:

According to the reviewers suggestion we completely adapted the manuscript accordingly using the term “macrosomia” for all children born with more than 4000g. Both models presented within the manuscript addressing the association between paraben exposure and birth weight were adjusted for the sex of the child, smoking during pregnancy, parental school education, gestational week at delivery, existence of siblings and age of the mother at birth.

We also carefully excluded the usage of „overweight/obese” in the context of birth weight measures.

COMMENT:

The authors define infancy as years 2-8. This is not correct as the age range corresponds to early-to mid-childhood. Infancy according to CDC is defined as the age period from birth-1 year of age.

Response:

We have rephrased the term infancy to ‘early to-mid-childhood’ throughout the manuscript.

COMMENT:

If the authors want to use a categorical outcome for rapid growth in infancy they could use the term “Rapid growth” defined as a Z score change of greater than 0.67 SD between birth and 6 or 12 months of age, and slow/average growth will be defined as a Z score change of equal or below 0.67 SD (Mendez et al. 2011; Monteiro and Victora 2005).

Response:

Thanks for that interesting suggestion. According to the causal context we would however stick to outcomes defined so far.

COMMENT:

The use of two-way ANOVA for the longitudinal analysis on prenatal paraben exposure and child BMI is not appropriate as it does not account for covariates and repeated outcomes. The authors should assess associations of prenatal exposures with longitudinal changes in BMI, using linear mixed effects models or similar techniques for the longitudinal repeated outcome measures. Potentially nonlinear functional forms of such associations should be examined by using flexible modeling techniques (e.g., splines).

Response:

We agree with the reviewer that the use of a two-way ANOVA test was misleading (as we didn’t intend this to be our main analysis but just wanted to display the raw data). Our main analysis model is an adjusted logistic regression model, comparing never overweight vs. ever overweight children between ages 2-8 years. However, as suggested we are now also including a longitudinal analysis. We applied a GEE model with an exchangeable correlation matrix to assess the effect of prenatal paraben exposure on BMI development between ages 1-8 years. The model was adjusted for sex, birth weight, age of mother, siblings and smoking during pregnancy. GEE is a population-level approach based on a quasi-likelihood function and provides the population-averaged estimates of the parameters. Based on model fit, we fit a cubic slope with 5 knot points at ages (12, 25, 38, 61, 98 moths) and also tested for

interactions with paraben exposure. The results are consistent with our logistic regression model showing no effects of methyl, ethyl and propyl paraben on BMI development. In contrast, there was evidence for a positive effect for iBuP and nBuP on BMI development, which was only seen in girls when stratifying for gender. The GEE model results can now be found in table 3 in the main manuscript. We excluded Supplementary Figure S2.

We thank again the reviewers for their valuable comments. Please, do not hesitate to contact us if there are still points to be clarified.

Thank you for taking the time to re-review our manuscript again.

Sincerely yours,

Tobias Polte, Beate Leppert, Kristin Junge, Irina Lehmann

REVIEWERS' COMMENTS:

Reviewer #4 (Remarks to the Author):

The revised manuscript is much improved. I have no further comments or concerns.

Reviewer #5 (Remarks to the Author):

I have no further comments or concerns.